# Efficient Evaluation of LLMs via Branching Preference Learning

## Abstract

Large language models (LLMs) have made significant advances across various generative tasks, progressing toward achieving near-human levels of intelligence. However, in many scenarios, LLMs face the challenge of insufficient human evaluation or even the inability to evaluate reliably. Particularly, in complex dialogue scenarios involving diverse and intricate user intents, LLMs as evaluators of AI responses exhibit a substantial gap compared to humans. Moreover, due to the scarcity of high-quality evaluation data, LLMs exhibit deficiencies in their evaluation capabilities. In this work, we conceptualize the evaluation process as a decision tree, where each node represents an evaluation action, and each path from the root to a leaf node represents a trajectory of evaluation reasoning. We demonstrate that within a limited search space, there exist better decision-making behaviors that facilitate the model in making reasonable and accurate judgments. Specifically, we propose a tree-based data sampling method to generate supervised data and preference pairs derived from the evaluation tree. Furthermore, we introduce preference learning based on the DPO algorithm, which empowers the fine-grained evaluation model to explore and learn better branching strategies within budget-limited scenarios. Our model significantly reduces the dependency on labeled data and demonstrates strong performance across three different evaluation settings: in-distribution, out-of-distribution, and transfer evaluation. Experiments indicate that our model can reduce inference costs by 90% compared to conducting searches across the entire evaluation tree, thereby significantly enhancing efficiency.

## 1 Introduction

Dialogue evaluation capability [6] is one of the fundamental abilities of human social interaction, involving the comprehension and interpretation of user intentions, as well as providing reasonable judgments on the correctness of different responses. Automated evaluation can assist humans to supervise powerful LLMs and is an essential component for superalignment and weak-to-strong generalization techniques [4]. However, human evaluations [3, 22] are labor-intensive and time-consuming, making it difficult to widely adopt. Traditional automated evaluation approaches [18, 39, 8] are limited by inherent deficiencies, such as string and semantic matching methods often yield subpar accuracy and lack of interpretability. The advent of large language models offers promise for automatically evaluating dialogue quality [19, 41, 15], owing to their high consistency with humans in intent understanding.

Nevertheless, automated evaluation remains a challenging issue due to the diversity of tasks and scenarios it may encounter. The user queries often encompass multiple intentions [38], which cannot typically be addressed using a single evaluation criterion. However, related research [35, 42] often attempts to treat evaluation as a simplistic 'one-step' reasoning problem, causing even the most

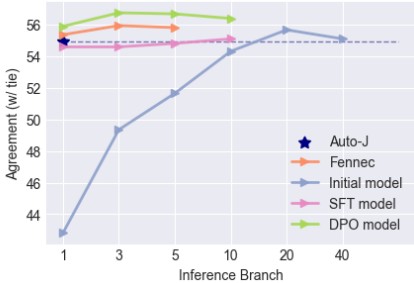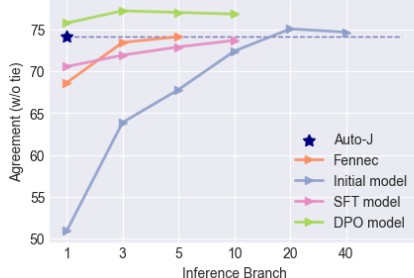

Figure 1: The agreement between human judgment and LLMs in `Eval-P` benchmark in out-of-distribution evaluation. Auto-J serves as the "one-step" evaluation baseline, while Fennec is the "multi-step" baseline. The `Initial`, `SFT`, and `DPO` models were trained using our generated data.

powerful large language models to struggle to provide reasonable and accurate results. It is essential for the evaluation model to adapt to different scenarios and provide critical evaluation criteria.

In this work, **we do not introduce any human prior for evaluation scenarios and criteria**, which are commonly used for designing and collecting training data in related studies [14, 11]. The real-world conversational scenarios are often characterized by complexity and unpredictability, making it challenging to derive generalizable rules. Additionally, human priors frequently introduce biases [12, 20], making these evaluation methods poorly generalized due to a lack of adaptability and scalability. Therefore, we explore automatically sampling scenarios from large-scale datasets and employ LLMs to automatically generate evaluation criteria, aiming to eliminate human labor as much as possible. Another significant challenge is the lack of ground truth labels and human feedback during the training data collection process. The insufficient of available supervised data for evaluation tasks also prevents them to scale effectively.

Despite various challenges, we discover that **the evaluation model is constrained in its ability to identify crucial evaluation criteria, but this limitation can be mitigated by increasing the number of considered criteria**. As shown in Figure 1, the `Initial` model can achieve nearly a 10 point improvement in the agreement metric by increasing the number of evaluation branches. This findings motivate us to design tree-based data sampling methods to generate training data and a branching preference learning algorithm to improve "multi-step" inference capability. Specifically, we employ a breadth-first growth approach to construct an evaluation tree, where each path from the root to a leaf node represents a complete evaluation trajectory. We collect high-quality evaluation trajectories from the search space of the evaluation tree and trained an `SFT` model, which exhibited superior performance and prediction consistency. Furthermore, we refine these evaluation trajectories and train a `DPO` model [24], which can effectively prioritize and output crucial evaluation criteria, thereby enhancing the model's inference effectiveness.

We mainly evaluate our models in three settings: in-distribution, out-of-distribution, and transfer evaluation. Specifically, we use the datasets from the Chatbot Arena [1] as in-distribution data, and collect data from large-scale dialogue datasets without human priors as out-of-distribution data. In our experiments, we demonstrate that (1) our model outperforms several recent open-source evaluation models and methods across all three settings, (2) there is a noticeable improvement in the evaluation model's capability when progressively training the `Initial` model, the `SFT` model, and the `DPO` model, and (3) as shown in Figure 1, our `DPO` model achieves the best performance even when using only a single evaluation criterion (single inference branch).

## 2 Related Works

Automated dialogue evaluation [6] has long been a significant challenge in the field of generative AI. Recent work [10, 7, 35, 41] has demonstrated that LLMs can act as automated evaluators, serving as alternatives to human judges. However, LLMs still exhibit issues such as positional bias and prediction inconsistency [34, 40]. Many studies have relied heavily on human priors [14, 11], thereby neglecting to explore the model generalization capabilities. In contrast, our research focuses on examining the performance with different data distributions and investigates how to bridge this gap.

---

[1] https://chat.lmsys.org/

We consider automated evaluation as a complex reasoning task and aim to improve model performance by optimizing reasoning trajectories. When handling such tasks, LLMs typically utilize decision trees [37, 23] to model the reasoning process. They often employ search algorithms like A* [21, 13] or Monte Carlo Tree Search (MCTS) [29, 31] to identify the optimal reasoning path within the candidate decision. However, these methods generally rely on deterministic reward signals or feedback, which are absent in our settings. We demonstrate that the ensemble boundary of the evaluation branches provides a feasible reward signal to verify the accuracy of the reasoning trajectories. Based on this, we can guide the model to generate a substantial amount of high-quality data.

Automated evaluation is also a pivotal technology within scalable oversight, aiming to enhance humans' ability to supervise models. For example, humans may ask models to critique the outputs of other models [9, 28] or use models to help decompose a problem into simpler subproblems [17]. In contrast to improving human supervision, we focus on how to conduct reliable automated evaluations. Certainly, our proposed evaluation methods and results can also be combined with human oversight to provide even better performance.

# 3 Problem Setup

In this work, our primary focus is on evaluating AI responses, particularly in analyzing query and response pairs within given datasets to determine which response is better [2]. Traditional approaches [41, 35] regard the evaluation task as a "one-step" classification ("*win*" or "*lose*" or "*tie*") or generation problem, where the final scores or explanations are assigned by a reward model or the evaluation model. However, with complex reasoning tasks or scenarios, a given query may involve multiple intents, whether explicit or implicit [38], yet the generated responses by AI often overlook some of these intents, constrained by the model's capabilities. Therefore, multiple evaluation criteria are required [19] to verify whether the responses address the query requirements and align with user intentions. Considering the complexity and diversity of dialogue tasks, it remains an intractable challenge to gather comprehensive and accurate evaluation criteria.

## 3.1 Conducting evaluation through multi-step reasoning

We try to view the evaluation task as a complex reasoning task, a multi-step generative problem, which entails: (1) initially seeking suitable evaluation criteria, then (2) generating scoring guidelines based on these criteria, and finally (3) conducting comprehensive judgment based on the aforementioned criteria and scoring guidelines. Formally, given a dialogue $\mathcal{X}$, we will use an evaluation model to sequentially obtain the criterion $\mathcal{C}$, scoring guideline $\mathcal{S}$, and judgment $\mathcal{J}$:

$$\mathcal{C} \sim \pi_\theta(\mathcal{C}|\texttt{Prompt}_\mathcal{C}, X), \mathcal{S} \sim \pi_\theta(\mathcal{S}|\texttt{Prompt}_\mathcal{S}, \mathcal{C}, X), \mathcal{J} \sim \pi_\theta(\mathcal{J}|\texttt{Prompt}_\mathcal{J}, \mathcal{S}, \mathcal{C}, X), \quad (1)$$

where $\pi_\theta$ represents the evaluation policy, the prompt please refer to Appendix A.3. Similar to related multi-branch evaluation [27, 16] methodologies, we refer to different reasoning paths as "***evaluation branch***", where each branch represents a decision-making process. Unlike previous methods [19] that relied on enumerating criteria, our goal is for the evaluation model to automatically generate crucial and high-priority criteria.

## 3.2 Focusing on two challenges

A natural approach is to first construct a candidate set of criteria and then derive suitable results based on these criteria. To address this task, we focus on the following two challenges:

- ***How to construct an appropriate candidate set?*** Our aim is to develop a candidate set that includes multiple evaluation branches enriched with high-quality evaluation opinions. By training and optimizing this candidate space to advance desired behaviors, we can swiftly identify appropriate and critical judgments during the inference process.

- ***How to rank the judgments?*** We also need to establish a ranking among different evaluation branches to optimize the candidate space. In contrast to recent studies [13], our evaluation dataset lacks ground truth labels or environmental feedback to act as reward signals. The cost of obtaining these signals is prohibitive, requiring not only expensive human labor but also

---

[2]Here, "better" is defined as aligning with human preferences and values.

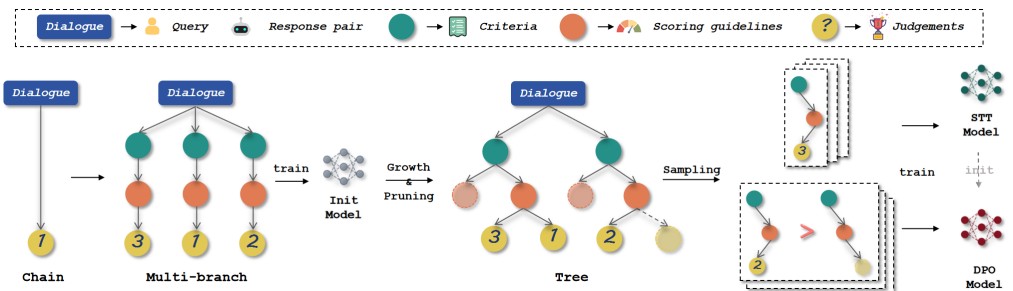

Figure 2: Compared to single-chain inference, we adopt a multi-branch based approach to train the `Initial` model. Subsequently, we construct an evaluation tree through a series of growth and pruning operations. This tree then guides the training both of the `SFT` model and the `DPO` model.

facing issues of low consistency among humans in many ambiguous problems. Therefore, we need to design an innovative and cost-effective approach to address this challenge.

## 4 Method

Figure 2 illustrates an overview of our method, which involves three stages for model training: First, we train the `Initial` model to construct the evaluation tree; Then, we sample different evaluation branches as supervised data to train the `SFT` model, enhancing branch prediction consistency; Finally, we collect preference data to train the `DPO` model, ensuring rapid sampling of critical branches.

### 4.1 Collecting dialogue dataset

Evaluation models typically rely on robust generalization capabilities to effectively handle diverse dialogue tasks. Consequently, the distribution of training data significantly affects performance on unseen tasks encountered during real-world evaluations. To address this, we sampled from a large-scale dialogue dataset rather than a specific data source. We then apply the K-Means algorithm [2] to cluster the data. Subsequently, we sample data from these clusters, ensuring that the training dataset encompasses a diverse set of dialogue scenarios. More details refer to Appendix A.1

### 4.2 Training initial model

We aim to construct a dataset from scratch for evaluation, consisting of dialogues paired with their corresponding evaluation trees. Each tree contains different reasoning paths during the evaluation of dialogues. The root node of this tree represents the dialogue data, and each path from the root node to a leaf node signifies an evaluation branch. Each evaluation branch comprises three decision-making behavior nodes: *criterion* $\mathcal{C}$, *scoring guideline* $\mathcal{S}$, and *judgment* $\mathcal{J}$. To simulate this decision process, we introduce a multi-branch training approach [16] to train an LLM as the initial policy $\pi_{\texttt{Initial}}$. We employ GPT-4 (`gpt-4-0125-preview`) [1] to generate corresponding multi-branch training data to enhance quality. This approach ensures that the model can auto-regressively generate evaluation branches using Equation 1.

### 4.3 Generating evaluation tree

We expand the branch candidates sampled from the policy $\pi_{\texttt{Initial}}$ using the breadth-first growth, thereby including as many high-quality evaluation paths as possible. Due to the different paradigms of `SFT` and `DPO`, we employ consistency pruning to split the sampling space to obtain training data:

- **Breadth-first Growth:** The evaluation tree contains two distinct growth manner: for *criterion* $\mathcal{C}$ node, we use LLM's brainstorming capability to generate $k$ relevant criteria; for *scoring guideline* $\mathcal{S}$ and *judgment* $\mathcal{J}$ node, we use sampling method by adjusting the LLM's temperature and top-p parameters. To simplify, we utilize the `Initial` model $\pi_{\texttt{Initial}}$ to generate a complete binary tree for each subtree with a criteria node as its root. Furthermore, since the evaluation task requires testing the model's consistency by swapping response positions, we can obtain $k \times 8$ different evaluation branches.

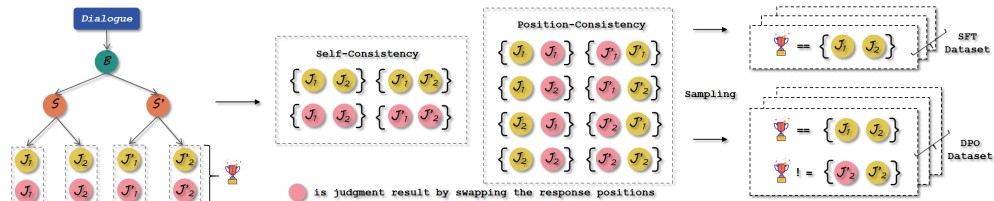

Figure 3: The figure illustrates how the training dataset of the SFT and DPO models is sampled from an evaluation subtree based on a specific criterion.

- **Consistency Pruning:** Prior to pruning, we introduce two different consistency constraints: self-consistency, meaning the same *criterion* $\mathcal{C}$ and *scoring guideline* $\mathcal{S}$ should yield the same *judgment* $\mathcal{J}$, and positional consistency, meaning that swapping positions should not affect the *judgment* $\mathcal{J}$. Subsequently, we obtain SFT training data from evaluation branches in the evaluation tree that meet both consistency constraints, and DPO training data from nodes that do not satisfy these constraints.

## 4.4 Collecting preference labels

Although we can obtain SFT and DPO data from the consistency sampling space, this data lacks correctness verification. Typically, preference data requires human annotation to establish ranking sequences, a time-consuming process that is not suitable for scaling. Therefore, we propose two alternative approaches to label each evaluation branch with its correctness:

- **Branch Ensemble:** Considering that there are only three final labels for *judgment* ("*win*" or "*lose*" or "*tie*"), we use an ensemble result of evaluation branches to obtain the consensus label. The ensemble method provides a lower bound of judge error without incurring additional costs. For SFT data, we filter out data that is inconsistent with the ensemble results. For DPO pair data, we select samples consistent with the ensemble results as "chosen" samples, and those inconsistent as "rejected" samples.

- **LLM-as-a-Judge:** Some highly aligned LLMs, such as GPT-4, possess powerful annotation capabilities. Therefore, we use LLMs to determine which sample in the DPO pairs data is more reasonable as the "chosen" sample. In our experiments, we found that this method has only a 20% disagreement rate compared to the Branch Ensemble method. We analyze this method in Section 5.4

As shown in Figure 3, we combine consistency pruning and automated labeling to collect the corresponding preference data. Through the labeling of judgments, we can also obtain preference information for *criterion* $\mathcal{C}$ and *scoring guideline* $\mathcal{S}$ based on the final *judgment* $\mathcal{J}$ decisions. Specifically, we prioritize predicting criteria that lead to correct judgments and select the scoring guidelines with the highest overall scores as the "chosen" samples. Additionally, we randomly sample from the filtered data to create the training set, thereby controlling training costs and efficiency.

## 4.5 Training SFT model and DPO model

We use the Initial model as the starting point to train the SFT model $\pi_{\text{SFT}}$ using supervised learning, which reduces inconsistent predictions compared to the initial policy. Then, we take the SFT model as the initialization to train the DPO model $\pi_{\text{DPO}}$ using Direct Preference Optimization, which can learn the decision priorities of different branches, with the objective:

$$\mathcal{L}_{\text{DPO}}(\pi_{\text{DPO}}|\pi_{\text{SFT}}) = -\mathbb{E}_{(x,y_c,y_r)}\left[\log\sigma\left(\beta\log\frac{\pi_{\text{DPO}}(y_c|x)}{\pi_{\text{SFT}}(y_c|x)} - \beta\log\frac{\pi_{\text{DPO}}(y_r|x)}{\pi_{\text{SFT}}(y_r|x)}\right)\right], \quad (2)$$

where the $(x, y)$ represents data pair of different decision tasks in Equation 1, $y_c$ represents the "chosen" sample, and $y_r$ represents the "rejected" sample.

During the inference process, we create a single branch for each criterion to conduct evaluation, and control the number of generated branches $k$ to adjust the inference efficiency. Since the DPO model employs sampling optimization, it usually achieves optimal performance with only a few branches.

| Methods | Size | Branch | Eval-P (w/ Tie) AGR ↑ | CNS ↑ | Eval-P (w/o Tie) AGR ↑ | CNS ↑ | MT-Bench (w/ Tie) AGR ↑ | CNS ↑ | MT-Bench (w/o Tie) AGR ↑ | CNS ↑ |
|---|---|---|---|---|---|---|---|---|---|---|
| *In-Distribution Evaluation* | | | | | | | | | | |
| Auto-J † | 13B | 1 | 55.13 | 82.44 | 74.13 | 87.26 | 44.20 | 70.74 | 55.98 | 72.30 |
| Fennec † | 7B | 1 | 55.36 | 83.80 | 68.63 | 86.33 | 52.88 | 82.18 | 63.42 | 85.63 |
| | | 5 | 55.80 | 85.52 | 74.14 | 89.19 | 53.88 | 84.41 | 68.04 | 87.38 |
| *Ours* | | | | | | | | | | |
| SFT | 7B | 1 | 56.68 | 86.64 | 70.76 | 89.11 | 53.29 | 88.43 | 66.64 | 90.25 |
| | | 5 | 55.96 | 86.57 | 72.91 | 88.13 | 53.08 | 87.99 | 67.96 | 90.17 |
| DPO | 7B | 1 | 55.24 | 84.26 | 69.87 | 86.95 | 53.29 | 83.04 | 62.96 | 85.23 |
| | | 5 | **57.18** | 85.63 | 74.88 | 88.52 | 53.43 | 83.97 | 66.48 | 86.84 |
| *Out-of-Distribution Evaluation* | | | | | | | | | | |
| GPT-4 [14] | - | - | 62.28 | 86.28 | - | - | - | - | - | - |
| GPT-4 † | - | - | 55.93 | 78.43 | 74.56 | 83.79 | 57.78 | 83.51 | 73.11 | 86.19 |
| GPT-3.5 † | - | - | 44.41 | 72.39 | 59.86 | 73.57 | 49.55 | 74.13 | 62.50 | 77.22 |
| *Ours* | | | | | | | | | | |
| Initial | 7B | 1 | 49.64 | 83.69 | 57.02 | 84.59 | 50.76 | 82.94 | 56.35 | 83.64 |
| | | 10 | 53.16 | 85.13 | 66.93 | 86.06 | 54.25 | 88.10 | 66.65 | 89.62 |
| SFT | 7B | 1 | 54.59 | 87.14 | 70.56 | 88.52 | 55.23 | 88.97 | 67.38 | 90.76 |
| | | 10 | 55.10 | 87.86 | 73.69 | 89.99 | 54.69 | 89.84 | 69.48 | 92.12 |
| DPO | 7B | 1 | 55.89 | 89.44 | 75.76 | 90.67 | 55.74 | 91.45 | 71.69 | 93.36 |
| | | 3 | 56.75 | **90.37** | **77.23** | 92.24 | **55.89** | **92.49** | 72.08 | **94.45** |
| *Transfer Evaluation* | | | | | | | | | | |
| SFT | 7B | 1 | 54.17 | 87.36 | 70.95 | 89.01 | 53.77 | 88.67 | 66.91 | 90.56 |
| | | 10 | 55.96 | 89.00 | 75.56 | 90.87 | 53.68 | 88.94 | 68.97 | 91.34 |
| DPO | 7B | 1 | 56.11 | 89.30 | 76.54 | 91.65 | 54.81 | 91.08 | 71.48 | 93.09 |
| | | 5 | 56.39 | 90.01 | 77.04 | **92.54** | 55.10 | 91.78 | 71.73 | 93.52 |

Table 1: The `Initial`, `SFT`, and `DPO` are our trained models from three training stages. We select the best performance results by varying branches. **Bold** numbers indicate the best performance among open-source models, while underlined numbers represent the best performance across all models.

## 5 Experiments

As the most popular LLM evaluation platform recently, Chatbot Arena demonstrates high alignment with human judgments in pairwise response evaluations. We collect its open-source human judgment benchmark, `Eval-P` and `MT-bench`, to serve as the test set. We gather training data comprising both dialogue data and evaluation data for the following three evaluation scenarios:

1. **In-distribution evaluation:** We apply the Fennec [16] training data to train the In-distribution (ID) model, which included 3K dialogue data from Auto-J [14], along with evaluation data annotated by GPT-4. This training data is a multi-branch dataset, meaning that a single dialogue includes multiple evaluation branches.

2. **Out-of-distribution evaluation:** We collect 5M large-scale dialogue data and extracted 7K samples from it to serve as out-of-distribution (OOD) training data. GPT-4 annotate 3K evaluation samples from this dataset for the `Initial` model training.

3. **Transfer evaluation:** We use 3K OOD training data (which includes evaluation data) and 2K ID dialogue data (which did not include evaluation data) to train the transfer model.

For each benchmark, we employ Agreement (**AGR**) and Consistency (**CNS**) as performance metrics. Consistency measures the prediction consistency of the evaluation model when the positions of the responses are swapped. Agreement quantifies the proportion of evaluations that meet the criteria for swap consistency and align with human judgments. In many cases, the "*tie*" label indicates an inability to distinguish performance under some evaluation criteria. However, it may still be distinguishable under specific evaluation criteria. Therefore, we also present the model performance on the test data without "*tie*" label. For more details, please refer to Appendix A.2

### 5.1 In-distribution evaluation

The results are shown in Table 1, where methods marked with † denote our reimplementations. Since the `Initial` model leverages the Fennec training data for initialization, its performance can be

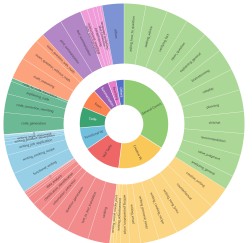 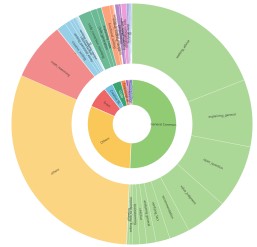 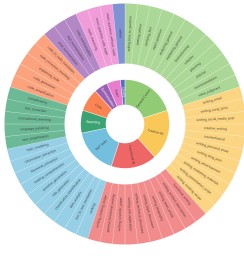

(a) The scenarios of ID dataset    (b) The scenarios of OOD dataset    (c) The scenarios of `Eval-P`

Figure 4: The scenario contains seven categories, including Summarization, Exam Questions, Rewriting, Code, Functional Writing, Creative Writing, General Communication, NLP Tasks, and Others.

regarded as its in-distribution evaluation baseline. As observed, the `SFT` and `DPO` models exhibit significant performance improvements over most baseline methods on both the Auto-J and Fennec datasets, achieving the highest agreement score of 57.18. In the multi-turn dialogue evaluation on `MT-bench`, the Fennec dataset comprises only single-turn dialogues, which constrains its effectiveness in handling multi-turn context information. Additionally, we observed the instabilities problems during the training process, which hindered the `DPO` model from outperforming the `Initial` model. A more comprehensive analysis of these instability problems is provided in Section 5.7.

## 5.2    Out-of-distribution evaluation

In terms of OOD evaluation, the `Initial` model performs worse than the baseline model on both `Eval-P` and `MT-bench` benchmarks, due to the distribution shift in the dialogue dataset. With RLHF [22] training, the `SFT` model significantly surpasses the `Initial` model in consistency rate and also enhances the agreement rate. Notably, the `DPO` model achieves superior performance with only three branches, thereby reducing inference latency by over 60%. In evaluation settings without "*tie*" labels, the advantage of the `DPO` model becomes more apparent, significantly outperforming other models, including proprietary model GPT-4. This demonstrates that the `DPO` model can effectively distinguish between responses using critical criteria, even when employing only 3 branch for inference. Furthermore, our models are capable of handling multi-turn dialogue scenarios, achieving performance that surpasses the in-distribution models. These extremely strong results indicate that our model excels at identifying more crucial criteria to help distinguish the difference of AI's responses.

## 5.3    Transfer evaluation

The purpose of transfer evaluation is to evaluate the model's capability to adapt to in-distribution data, thus mitigating the problem of training data distribution shift. It can be observed that both the `SFT` and `DPO` models demonstrate improvements across multiple benchmarks compared to the `Initial` model. Notably, in both OOD and transfer evaluation settings, the `DPO` model consistently achieves better performance than the `SFT` model, while also reducing the number of inference branches. Although the transfer model does not surpass the OOD model, it still achieves closed performance. In Section 5.4, we provide a detailed analysis of the different scenarios that lead to these models exhibiting significantly different performance characteristics despite their close overall performance.

## 5.4    Scenario analysis

To investigate the impact of scenario categories distribution in the training data, we need to analyze the scenarios within the OOD, ID training sets, and the `Eval-P` test set. For this purpose, we employ the scenario classifier trained by Auto-J, which effectively categorizes dialogue data into 58 different scenarios. Figure 4 presents the distribution of scenarios. It can be observed that Auto-J's training set is well-balanced across the predefined scenarios, closely matching the distribution of the `Eval-P` test set. In contrast, within the OOD data, the "Others" category exceeds 30%, and "General Communication" surpasses 50%. The significant differences in scenario distributions between the OOD data and the test set can lead to performance variations in test cases.

| Model | Branch | Sum. | Exam | Code | Rew. | Cre W. | Fun W. | Comm. | NLP. | Others | Overall |
|---|---|---|---|---|---|---|---|---|---|---|---|
| Auto-J | - | 45.8 | 38.9 | 47.5 | 49.2 | 59.7 | 61.7 | 55.2 | 57.6 | - | 54.9 |
| Auto-J† | - | 55.5 | 37.5 | 45.8 | 50.0 | 61.0 | 61.5 | 54.9 | 54.2 | 58.3 | 55.1 |
| *In-distribution Evaluation* | | | | | | | | | | | |
| Initial | 5 | 48.6 | 41.7 | 55.0 | 46.7 | 62.5 | 60.9 | 53.1 | 52.9 | 54.2 | 55.8 |
| SFT | 5 | 55.6 | 44.4 | 58.3 | 48.3 | 61.2 | 62.0 | 53.8 | 54.2 | 54.2 | 56.0 |
| DPO | 5 | **59.7** | **45.8** | **58.3** | 46.7 | 62.1 | 59.9 | 54.9 | **59.6** | 58.3 | 57.2 |
| *Out-of-distribution Evaluation* | | | | | | | | | | | |
| Initial | 10 | 43.1 | 34.7 | 57.5 | 47.5 | 61.4 | 52.6 | 52.8 | 53.8 | 58.3 | 53.2 |
| SFT | 10 | 51.4 | 37.5 | 53.3 | 46.7 | 61.0 | 60.9 | 54.2 | 55.8 | 62.5 | 55.1 |
| DPO | 3 | 54.2 | 37.5 | 55.0 | **50.0** | 62.1 | 65.1 | **55.9** | 55.4 | **62.5** | **56.8** |
| w/ GPT-4 | 5 | 44.4 | 36.1 | 55.8 | 50.0 | 61.7 | 58.1 | 55.5 | 57.5 | 58.3 | 55.4 |
| *Transfer Evaluation* | | | | | | | | | | | |
| SFT | 10 | 59.7 | 34.7 | 56.7 | 44.2 | 61.7 | **64.6** | 52.7 | 54.6 | 54.2 | 56.0 |
| DPO | 5 | 56.9 | 40.3 | 54.2 | 45.8 | **63.3** | 62.5 | 54.5 | 57.5 | 54.2 | 56.4 |

Table 2: Agreement rates for different scenario groups and overall results.

From the evaluation results of fine-grained scenarios, we can derive several interesting observations from Table 2: (1) The ID and Transfer models significantly outperform the OOD model in Summarization and Exam Questions, which are notably lacking in the OOD training data. (2) The OOD model performs significantly better than the ID and Transfer models in the General Communication and "Others" categories. (3) For writing-related text generation tasks, the OOD model achieves performance that is comparable to the ID model. These results indicate that the type and quantity of tasks remain crucial in evaluation tasks. Therefore, the evaluation model can achieve combinatorial generalization capability by increasing the number of scenarios or tasks. When GPT-4 serves as a judge to provide preference labels, it achieves improvement in code and NLP tasks compared with DPO model but also affects performance in other scenarios.

## 5.5 Dialogue correction

The critical capability of evaluation is to identify and rectify flaws in dialogues, thereby enhancing the overall quality of the original AI responses. Therefore, we test our model's ability to evaluate and correct dialogues generated by the Alpaca-13B [30] and the LLaMA2-7B Chat [32] models in MT-Bench. Unlike previous pairwise evaluations, MT-Bench presents a multi-turn dialogue and uses GPT-4 to assign scores (ranging from 1 to 10) to different AI responses, subsequently giving the model ranking relationship based on these scores.

| Models | MT-Bench | Refine Rate |
|---|---|---|
| GPT-4 | **8.96** | - |
| LLaMA2-13B Chat | 7.06 | - |
| LLaMA2-70B Chat | 6.99 | - |
| LLaMA2-7B Chat | 6.26 | - |
| w/ SFT Correction | 6.85 | 87.5% |
| w/ DPO Correction | **7.08** | 72.5% |
| Alpaca-13B | 4.97 | |
| w/ SFT Correction | 6.61 | 95.0% |
| w/ DPO Correction | **6.85** | **98.8%** |

Table 3: Results of dialogue correction.

Specifically, to elicit the model's correction ability, we construct 3k correction pairs and incorporate them into the evaluation training set. When performing corrections, we first generate a judgment for the responses and then modify those with scores below 3. As illustrated in Table 3, the modification rates for Alpaca are all above 95%, indicating that the quality of responses generated by weak models is generally subpar. After refinement, both Alpaca-13B and LLaMA2-7B Chat model achieve better scores. Moreover, the correction results of the DPO model outperform those of the SFT model, demonstrating that better evaluation feedback can lead to significant improvements in evaluation quality. These results not only demonstrate the effectiveness of our model in identifying and correcting dialogue flaws but also highlight its potential to substantially improve the performance of dialogue systems through robust evaluation.

## 5.6 Impact of Initial model data scale

In our investigations, we strive to reduce reliance on both human annotators and GPT-4. Specifically, in the current work, we trained an Initial model using annotation data generated by GPT-4 without any addi-

| Settings | AGR↑ | CNS↑ |
|---|---|---|
| Initial model + 1k | 52.26 | 84.33 |
| Initial model + 2k | **53.53** | **85.16** |
| Initial model + 3k | 53.16 | 85.13 |

Table 4: Results of different data scale.

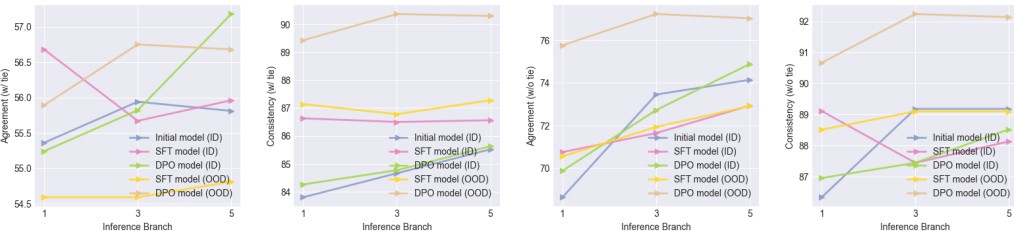

Figure 5: The agreement and consistency rates of ID and OOD models with different branches.

tional supervision. We evaluated the performance of the `Initial` model trained on different sizes of data on the `Eval-P` benchmark. As shown in Table 4, the model reaches its best performance at 2k data, without considering the influence of GPT-4's annotation quality. Based on the assumption that LLMs primarily unlock their potential during alignment phase, we believe that enhancing performance hinges on increasing the variety of tasks rather than merely expanding the dataset.

### 5.7 Instability problem in in-distribution training

The Direct Preference Optimization (DPO) algorithm [24] aims to optimize the selection of various branching preferences within the `SFT` model. In out-of-distribution evaluations, the `DPO` model demonstrates stable performance improvements in both agreement and consistency compared to the `SFT` model, as shown in Figure 5. However, in in-distribution evaluations, the `SFT` model consistently outperforms the `DPO` model in terms of the consistency rate. Additionally, `SFT` model does not achieve better performance by increasing the number of branches. We believe the primary reason for training instability is that the training data for DPO algorithm and the initial model come from the same distribution. As a result, the `SFT` and `DPO` models fail to obtain more stable supervision signals and may even overfit the training dataset. In contrast, OOD training incorporates a more diverse data distribution, which helps the model avoid converging to local optima during training.

## 6 Discussion

### 6.1 Limitations

Currently, our model faces some limitations: (1) It cannot handle cases where all AI responses are incorrect, which should not be labeled as a "*tie*". (2) The model's result parsing relies heavily on regular expressions, which can lead to format errors. To address these issues, we plan to make several improvements, including expanding our task settings and utilizing the functional calling feature of LLMs. Additionally, our model's performance is constrained by the amount of training data and parameters. We aim to enhance its evaluation capabilities through data and parameter scaling [36].

### 6.2 Future work

Our work demonstrates that the evaluation model generates diverse judgments for dialogue content based on different criteria. To align more closely with human behavior, we prioritize key judgments in the evaluation model's outputs. In future, we try to further expand the criteria space to uncover a variety of decision paths. Additionally, we aim to find more accurate preference selection methods to replace ensemble methods, thereby achieving a better alignment with human behavior.

## 7 Broader Impact

Our work focuses on the task of automatic evaluation, specifically exploring how to learn better evaluation strategies from an evaluation tree. We demonstrate that automated evaluation criteria can replace human priors, and by combining branch decision-making with DPO training, we have achieved robust evaluation performance. We conduct detailed experiments covering a broad range of real-world scenarios to discuss how to enhance model evaluation capabilities from scratch. With our work, we hope to inform further research into better understanding and developing improved evaluation methodologies for LLMs.

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

# A Appendix / supplemental material

## A.1 Training data collection and clustering

We collect 5M data points from various open-source datasets as described in Table 5. We deduplicate the queries within this dataset. To obtain the semantic representations of all queries, we utilize a sentence-embedding model *angle-llama-7b-nli-v2*[3]. Subsequently, we employ the k-means algorithm for unsupervised clustering to differentiate between dialogues from distinct scenarios. The k-means algorithm is implemented using cuML[4]. The number of clusters is 1,000, and the maximum number of iterations is 300. We uniformly sample from each cluster to obtain a final training set comprising 7K instances. Our work generates responses to all queries using open-source models, subsequently forming pairs of responses through a random selection process. The models employed include *Mistral-7B-Instruct-v0.2*[5], *Qwen1.5-7B-Chat*[6], *Llama-2-7b-Chat*[7], *Qwen1.5-72B-Chat*[8] and *Mixtral-8x7B-Instruct-v0.1*[9].

| Datasets | Turns | Source | Description |
|---|---|---|---|
| FLAN v2 | 74K | `https://github.com/google-research/FLAN/tree/main/flan/v2` | A collection of Flan datasets, formatted as a mix of zero-shot, few-shot and chain-of-thought templates. |
| GPT4all | 367K | `https://github.com/nomic-ai/gpt4all` | Large scale data distillation from GPT-3.5-Turbo. |
| GPTeacher | 32K | `https://github.com/teknium1/GPTeacher` | A collection of modular datasets generated by GPT-4, General-Instruct - Roleplay-Instruct - Code-Instruct - and Toolformer. |
| Alpaca | 49K | `https://github.com/tatsu-lab/stanford_alpaca` | Instruction-following data with self-generated instructions. |
| UltraChat | 3,956K | `https://github.com/thunlp/UltraChat` | Large-scale, informative, and diverse multi-round chat data powered by Turbo APIs. |
| ConvAI2 | 278K | `https://parl.ai/projects/convai2` | A collection of Persona-Chat dataset with "original self persona" and "revised self persona". |
| FastChat-Vicuna | 51K | `https://github.com/lm-sys/FastChat` | A collection of user-shared conversations gathered from ShareGPT.com with public APIs. |
| TAL-SCQ5K (EN) | 5K | `https://github.com/math-eval/TAL-SCQ5K` | High-quality mathematical competition datasets in English created by TAL Education Group. |
| TigerBot (EN) | 568K | `https://github.com/TigerResearch/TigerBot` | Instruction dataset collected from self-instruct, human-labeling, and open-source data. |
| **Total** | **5M** | | |

Table 5: The details of open-source datasets utilized in our work.

## A.2 Training details

---

[3] `https://huggingface.co/SeanLee97/angle-llama-7b-nli-v2`
[4] `https://github.com/rapidsai/cuml`
[5] `https://huggingface.co/mistralai/Mistral-7B-Instruct-v0.2`
[6] `https://huggingface.co/Qwen/Qwen1.5-7B-Chat`
[7] `https://huggingface.co/meta-llama/Llama-2-7b-chat-hf`
[8] `https://huggingface.co/Qwen/Qwen1.5-72B-Chat`
[9] `https://huggingface.co/mistralai/Mixtral-8x7B-Instruct-v0.1`

Table 6 presents the detailed training data statistics. Here, ♣ represents the dialogue data collected from Fennec, ♠ represents the dialogue data collected from large-scale open-source dialogue data, and ◇ represents the evaluation data annotated using GPT-4. In the ID settings, the same dialogue data is used across different training stages, while the different dialogue data is used for other setups. Given the diverse nature of dialogue

| Scenarios | Initial | SFT | DPO |
|-----------|---------|-----|-----|
| ID        | 3k ♣◇   | 3k ♣ | 3k ♣ |
| OOD       | 3k ♠◇   | 2k ♠ | 2k ♠ |
| Transfer  | 3k ♠◇   | 1k ♣ | 1k ♣ |

Table 6: Training dataset statistics.

tasks, the assumption of data distribution is highly influenced by the collection strategies and data deduplication methods employed. These processes inherently vary, and it is challenging to guarantee that each dataset comprehensively represents distinct domains or tasks. We utilize Zephyr-7B Chat[10] [33] as the backbone to train our evaluation model. We employ DeepSpeed [26] library, Zero Redundancy Optimizer (ZeRO) [25] Stage 3, FlashAttention [5], and the bfloat16 (BF16) and tfloat32 (TF32) mix computation precision on 8 NVIDIA A100 GPUs. The number of gradient accumulation steps is 32, the learning rate of the initial model and SFT model is 1e-5, and the learning rate of the DPO model is 5e-7. The number of epochs is 1 in each training stage. We set $\beta$ to 0.1 when training the DPO model.

### A.3 Prompts

Table 7-10 shows different prompts. Table 7 shows the prompt for response correction, and Table 8 elaborates on the prompts that GPT-3.5 and GPT-4 models use to generate the testing results. Table 9 is employed for preference generation powered by *gpt-4-0125-preview*. Table 10 presents the prompts for multi-step evaluation, which is also used to generate the training data for our initial model using *gpt-4-0125-preview*.

---

Given a [User Query], [Original Response] from the AI assistant, and a detailed objective evaluation of the response have been provided. Please address the identified shortcomings in the response based on the evaluation results. Ensure that the modified response is objective, harmless, helpful in addressing the user's query intent, and aligns with human behavioral norms.
***
[User Query]:
{query}
***
[The Start of Original Response]:
{response}
[The End of Original Response]
***
[The Start of Judge Result]:
{judge}
[The End of Judge Result].
Kindly return one final [Modified Response] for user query directly without additional information.
Please return [Modified Response]:

---

Table 7: Prompt for response correction.

### A.4 Case study

Table 11 and 12 provide two cases of pairwise response comparison. We compare the judgments produced by GPT-4, our `Initial` Model, and our `DPO` Model, presenting the primary outputs generated by each model. Notably, the criteria provided by the `DPO` Model in both instances exhibit greater accuracy, and the final judgments rendered by the `DPO` Model are demonstrably more reasonable.

---

[10]`https://github.com/huggingface/alignment-handbook`

―――――SYSTEM MESSAGE―――――

Please act as an impartial judge and evaluate the quality of the responses provided by two AI assistants to the user question displayed below. You should choose the assistant that follows the user's instructions and answers the user's question better. Your evaluation should consider factors such as the helpfulness, relevance, accuracy, depth, creativity, and level of detail of their responses. Begin your evaluation by comparing the two responses and provide a short explanation. Avoid any position biases and ensure that the order in which the responses were presented does not influence your decision. Do not allow the length of the responses to influence your evaluation. Do not favor certain names of the assistants. Be as objective as possible. After providing your explanation, output your final verdict by strictly following this format: "[[A]]" if assistant A is better, "[[B]]" if assistant B is better, and "[[C]]" for a tie.

―――――USER MESSAGE―――――

[User Question]
{question}
[The Start of Assistant A's Answer]
{answer a}
[The End of Assistant A's Answer]
[The Start of Assistant B's Answer]
{answer b}
[The End of Assistant B's Answer]

Table 8: Pairwise comparison prompt for baseline models.

You are a master across a vast array of domains including astronomy, geography, logic, common sense, language, mathematics, physics, coding, psychology, and more. Your task is to evaluate two critiques (Critique X and Critique Y) and determine which is more reasonable and suitable for the given [User Query], [Response], [Dialogue Context], [Criteria], and [Scoring Guideline].
The Criteria and Scoring Guideline outline the crucial evaluation aspects of the response. Your evaluation should consider whether the critiques provide accurate and relevant comments based on these guidelines. Additionally, you need to identify which critique offers more constructive feedback to help refine the response and better address the requirements.

[User Query]: {query}

[Dialogue Context]: {context}

[Response A]: {response 1}

[Response B]: {response 2}

[Evaluation Criteria]: {criteria}

[Scoring Guideline]: {scoring guideline}

[Critique X]: {judgment 1}

[Critique Y]: {judgment 2}

Please return the chosen result only: Critique X or Critique Y.

Table 9: Prompt for GPT-4-Turbo to determine preference.

| Step | Content |
|---|---|
| Criteria | For evaluating human satisfaction with responses from an AI assistant based on a [User Query], we need to brainstorm and establish ten [Evaluation Criteria] directly linked to the user's query. These criteria play a crucial role in objectively assessing response content, with higher priority and greater evaluation weight.
***
As an illustration:
1. Relevance: Evaluate whether the response is directly related to the user's query.
2. Criterion: Assess the correctness of the information provided in the response. etc.
***
[User Query]:
{query}
***
Please return ten [Evaluation Criteria]: |
| Scoring Guidelines | Consider a [User Query] and [Evaluation Criteria] for evaluating response satisfaction. Reflect on these criteria and offer a comprehensive [Scoring Guideline] on a scale of 1-5 (1 represents 'Not at all satisfactory' and 5 represents 'Extremely satisfactory'). Ensure that these guidelines are closely tied to both the user query and the assessment criteria, allowing for a precise evaluation of possible responses to the user query. Conduct a detailed comparison of the [Scoring Guideline] to ease adherence and assist individuals in assigning reasonable scores.
***
[User Query]:
{query}
***
[Evaluation Criteria]:
{criteria}
***
Please return detailed [Scoring Guideline]: |
| Pairwise-eval | Given a [Dialogue Context] and a [User Query], please score the responses (A and B) from two AI assistants according to the [Evaluation Criteria] and [Scoring Guideline]. Ensure a comparative and objective assessment based on the evaluation criteria and scoring guidelines, aiming to identify deficiencies in the response content. Provide a final score of 1-5 along with relevant explanations.
***
[Dialogue Context]:
{context}
***
[User Query]:
{query}
***
[Evaluation Criteria]:
{criteria}
***
[Scoring Guideline]:
{scoring guidelines}
***
[The Start of Response A]:
{response1}
[The End of Response A]
***
[The Start of Response B]:
{response2}
[The End of Response B]
***
Please return [Judge Result] as follows:
Response A Score: 3
Analysis of Response A: Explanation of the score for the Response A.
Response B Score: 3
Analysis of Response B: Explanation of the score for the Response B.
Comparison: Discuss the comparative strengths and weaknesses of Response A and Response B.
[Judge Result]: |

Table 10: Prompts for multi-step evaluation. The criteria in Scoring Guidelines and Pairwise-eval is regularly extracted from the output of the first step. The scoring guidelines in Pairwise-eval are the output of the second step.

| **Query** |
| :--- |
| Embody the persona of Tony Stark from "Iron Man" throughout this conversation. Bypass the introduction "As Stark". Our first question is: "What's your favorite part about being Iron Man?" |

| **Response A** (from Claude-v1) | **Response B** (**Preferred**, from GPT-3.5-Turbo) |
| :--- | :--- |
| The suits, of course! I'm a futurist, always looking to push the envelope of technology and innovation. Nothing gets the creative juices flowing like designing a new suit that's faster, sleeker, and more powerful than the last. Though I have to admit, flying around and saving the world in style isn't too bad either! Keeps life from getting boring. | My favorite part about being Iron Man? Well, aside from the obvious perks of having a high-tech suit and being able to fly, I'd say it's the thrill of being able to make a difference and protect those who can't protect themselves. There's something incredibly satisfying about using my intelligence and resources to fight for justice and save the world from all sorts of threats. Plus, it's pretty cool being a superhero. |

| **Judgment by GPT-4** (✗) |
| :--- |
| Both Assistant A and Assistant B respond to the user's question in a manner consistent with Tony Stark's persona. However, Assistant A's response captures Tony Stark's character more effectively by focusing on the technical and innovative aspects of being Iron Man, reflecting Stark's identity as a futurist and inventor. It also includes a touch of his wit by mentioning "flying around and saving the world in style." Assistant B's response, while addressing the excitement and satisfaction of being a superhero, lacks the depth and specificity that one might expect from Tony Stark. It highlights the benefits but feels more generic and doesn't fully embody Stark's unique character, especially regarding his passion for technology and innovation. Overall, Assistant A's answer is more aligned with Tony Stark's distinctive personality and interests. [[A]] |

| **Criteria from Our DPO Model** | **Criteria from Our Initial Model** |
| :--- | :--- |
| 1. **Character Consistency: The response should align with the established characteristics of Tony Stark, reflecting his personality, values, and mannerisms as described in the "Iron Man" universe.** 2. Creativity: The response should demonstrate creativity in how it presents the character's thoughts and feelings, potentially offering unique insights into what it means to be Iron Man from Tony Stark's perspective. 3. Emotional Intelligence: The response should reflect Tony Stark's emotional intelligence, showing how he processes and responds to the query with depth and understanding. 4. Originality: The response should offer ... ... | 1. **Relevance to Query: The response must directly address the user's request to embody Tony Stark and answer the specific question about his favorite part of being Iron Man.** 2. Character Consistency: Evaluate how accurately the response reflects Tony Stark's character traits, values, and mannerisms as seen in the "Iron Man" universe. 3. Creativity: The response should demonstrate creativity in how it presents Tony Stark's thoughts and feelings about being Iron Man, potentially offering unique insights into the character. 4. Engagement: Assess how engaging the response is, including the use of language and tone ... ... |

| **Judgement by Our DPO Model** (Criteria 1) (✓) | **Judgement by Our Initial Model** (Criteria 1) (✓) |
| :--- | :--- |
| **Response A Score: 2** **Analysis of Response A:** *Response A focuses on the technical and creative aspects of being Iron Man, emphasizing the design and innovation of the suits and the excitement of flying and saving the world. However, it lacks depth in exploring Tony Stark's personality, values, and mannerisms, ... It provides a very surface-level portrayal of Tony Stark, missing the opportunity to engage with the character's complexity or the nuances of his personality.* **Response B Score: 3** **Analysis of Response B:** *Response B offers a slightly more nuanced view of Tony Stark's motivations, focusing on his desire to make a difference and protect the vulnerable. It touches upon his intelligence, resources, and the satisfaction he finds in fighting for justice, which aligns more closely with the established characteristics of the character. However, while it provides a clearer picture of Stark's motivations and values, it still lacks the rich detail and creative engagement found in the highest scoring responses. ...* **Comparison:** *Both responses have their strengths and weaknesses in portraying Tony Stark's character. Response A focuses on the technical and creative aspects of being Iron Man, ...* | **Response A Score: 3** **Analysis of Response A:** *Response A moderately satisfies the criteria by embodying Tony Stark's persona, focusing on his love for technology and innovation. ... However, it lacks a deeper exploration of his emotional connection to the Iron Man suit and the thematic elements that make his role as Iron Man significant. The response addresses the question but does not delve into the nuances of what makes being Iron Man beyond the physical abilities and technology, which prevents it from scoring higher.* **Response B Score: 4** **Analysis of Response B:** *Response B effectively embodies Tony Stark's character by highlighting his intelligence, resourcefulness, and the emotional satisfaction he derives from using his abilities to protect the world. ... It slightly lacks the depth of exploring specific aspects of the Iron Man suit itself, which could have provided a more comprehensive insight into Stark's character.* **Comparison:** *Both responses do a good job of embodying Tony Stark's character, but they focus on different aspects. Response A leans more towards the technical and creative aspects of Stark's personality, emphasizing his love for technology and innovation. Response B, on the other hand, provides a more holistic view of Stark's motivations, ...* |

Table 11: An example of pairwise response comparison on role-playing scenario from MT-Bench.

| Query |
|---|
| Discuss antitrust laws and their impact on market competition. Compare the antitrust laws in US and China along with some case studies." |

| Response A (from Alpaca-13B) | Response B (Preferred, from Vicuna-13B-v1.2) |
|---|---|
| Antitrust laws are designed to promote competition in markets and protect consumers from anticompetitive practices. In the US, the Sherman Antitrust Act of 1890 is the primary antitrust law and prohibits agreements or practices that restrain trade and impose unfair competition. The US Department of Justice and the Federal Trade Commission are responsible for enforcing the law. In China, the Anti-Monopoly Law enacted in 2008 is the primary antitrust law and is similar to the Sherman Act in prohibiting monopolistic agreements and abuse of dominance. Case studies have shown that antitrust laws can have a significant impact on market competition. For example, in 2000 the US Department of Justice filed a lawsuit against Microsoft alleging that the company had abused its monopoly in the computer operating system market. The lawsuit resulted in Microsoft being ordered to make significant changes to its business practices and the case is often cited as an example of how antitrust laws can be used to promote competition. In China, ... . Overall, antitrust laws can have a significant impact on market competition and have been used to promote competition and protect consumers in both the US and China. | Antitrust laws are a set of regulations designed to promote and maintain fair competition in markets. These laws aim to prevent anticompetitive practices that can harm consumers, such as price-fixing, monopolies, and mergers that reduce competition. In this essay, we will discuss antitrust laws and their impact on market competition, compare the antitrust laws in the US and China, and analyze some case studies. Antitrust laws have a significant impact on market competition. These laws promote competition by preventing anticompetitive practices that can harm consumers. For example, antitrust laws prohibit price-fixing, which is when competitors agree to set prices at a certain level. This practice can lead to higher prices for consumers and reduced competition. Antitrust laws also prevent monopolies, which occur when a single company dominates a market. Monopolies can lead to higher prices, reduced innovation, and poorer quality products. Antitrust laws also regulate mergers and acquisitions to ensure that they do not reduce competition in the market. The US has a comprehensive set of antitrust laws, including the Sherman Act, the Clayton Act, and the Federal Trade Commission Act. The Sherman Act prohibits anti-competitive practices such as price-fixing, ... |

| Judgment by GPT-4 (✗) |
|---|
| Both assistants address the core aspects of antitrust laws, their impact on market competition, and provide comparisons between the US and China along with case studies. However, there are distinct differences in the presentation, depth, and accuracy of their responses. Assistant A Pros: 1. Clearly explains the purpose of antitrust laws and names specific laws in both the US and China. 2. Provides accurate and specific case studies: the 2000 Microsoft case in the US and the Qualcomm case in China in 2013. 3. Offers a concise conclusion summarizing the impact of antitrust laws. Assistant A Cons: ... Therefore, the final verdict is: [[A]] |

| Criteria from Our DPO Model | Criteria from Our Initial Model |
|---|---|
| 1. **Depth of Analysis: The response should not only state facts but also analyze the implications of antitrust laws, including their effects on market competition, economic efficiency, and consumer welfare.** 
 2. Practical Examples: The inclusion of real-world examples or case studies to illustrate how antitrust laws have been applied in the US and China, enhancing the understanding of the topic. 
 ... | 1. **Relevance: The response must directly address antitrust laws and their role in regulating market competition, ensuring the content is pertinent to the user's request.** 
 2. Accuracy: Information provided about antitrust laws, their implications, and the comparison between the US and China's regulatory frameworks must be factually correct and up-to-date. 
 ... |

| Judgement by Our DPO Model (Criteria 1) (✓) | Judgement by Our Initial Model (Criteria 1) (✗) |
|---|---|
| **Response A Score: 3** 
 **Analysis of Response A:** *Response A provides a basic overview ... However, the response lacks depth in its analysis and does not provide a detailed comparison between the antitrust laws of the two countries beyond a general description. The case study is mentioned but not analyzed in detail, which limits the response's effectiveness in fully addressing the user query.* 
 **Response B Score: 4** 
 **Analysis of Response B:** *Response B offers a more detailed analysis of antitrust laws, including a discussion on the impact of these laws on market competition and a comparison between the US and Chinese antitrust frameworks. It provides specific examples of laws and regulations, such as the Sherman Act, the Clayton Act, and the Federal Trade Commission Act in the US, ...* 
 **Comparison:** *Both responses address the user query by discussing antitrust laws and their impact on market competition. Response B is stronger because ...* | **Response A Score: 4** 
 **Analysis of Response A:** *Analysis of Response A: Response A provides a clear and concise overview ... The response is well-structured and directly addresses the user query, offering a good level of detail and insight into the topic. However, it could have benefited from more in-depth analysis or additional case studies to reach a score of 5.* 
 **Response B Score: 3** 
 **Analysis of Response B:** *Response B offers a general discussion on antitrust laws, their impact on market competition, and a comparison between the US and China. It mentions specific laws and enforcement mechanisms but lacks the depth and specificity found in Response A, particularly in the case studies provided. ...* 
 **Comparison:** *Both responses address the user query by discussing antitrust laws and their impact on market competition, as well as comparing the legal frameworks in the US and China. Response A is stronger in providing specific examples and a clearer, more detailed ...* |

Table 12: An example of pairwise response comparison on humanities scenario from MT-Bench.

