# OpenReview forum: "Efficient Evaluation of LLMs via Branching Preference Learning"
_NeurIPS.cc/2024/Conference — Submitted to NeurIPS 2024_

### Official Review · Reviewer_RFGW · 2024-06-30

**Soundness:** 2
**Presentation:** 1
**Contribution:** 2
**Rating:** 4
**Confidence:** 5

**Summary:**

In this work, the authors conceptualize the evaluation process as a decision tree, where each node represents an evaluation action, and each path from the root to a leaf node represents a trajectory of evaluation reasoning. The authors demonstrate that within a limited search space, there exist better decision-making behaviors that facilitate the model in making reasonable and accurate judgments.

**Strengths:**

1. The idea of branching LLM evaluation is interesting and novel.

**Weaknesses:**

1. The authors missed a lot of key related works, including close-ended benchmarks such as MMLU, MMLU-pro, MixEval, GSM8k, GSM1k, etc; open-ended benchmarks such as Arena-Hard, AlpacaEval, WildBench, Chatbot Arena, etc.
2. I think the writing needs improvement. Now it's not easy for a reader to get what you are focusing on. If you are doing evaluation, then try to use some pipeline figures and comprehensive captions to describe the core idea. Besides, all captions in this paper is misleading, not telling the reader about what is happening in the table or figure; also, there lacks some key sections such as conclusion.
3. How to measure the quality of the proposed evaluation? I think just evaluating 5-6 models is far from enough. Beyond that, how is the model rankings related with Chatbot Arena or some other popular benchmarks such as MMLU?

**Questions:**

Is it necessary to consider evaluations specifically for dialogue settings? I think all LLM evaluations are in a dialogue settings, the difference is just one-turn or multi-turn evals. Just calling it open-ended eval will be better aligned with the LLM research community.

**Limitations:**

Yes

---

> ### Author Rebuttal · Authors · 2024-08-06
>
> Thanks for your review comments.
>
> **For Concerns**:
>
> > Q: The authors missed a lot of key related works
>
> 1. We clarify our task and related work (refer to common responses 1, 2, and 4 for more details):
>
> - Our work has broader application scenarios; it is not only for benchmarking model capabilities but also for reward modeling, preference data construction, and correction models [1] (see Sec 5.5 for related experiments). In subsequent experiments, we show our results on Chatbot Arena, which exhibit a strong correlation with human scoring.
>
> - Our work supplements benchmarks like MMLU and GSM8k (as discussed in common responses 1 and 2). While MMLU and similar evaluation tests focus on correctness scoring, many real-world problems don't have strict binary labels (correct or incorrect) and require preference scoring. Our research aims to address the shortcomings of these automated evaluation benchmarks.
>
> - Our work can provide an open-source, automated alternative to replace GPT-4 for evaluations (Like WildBench relies on GPT-4 for evaluation).
>
> - Our contributions (see common response 4 for more details): our aim is not only to provide a better evaluation method for ranking a large number of LLMs but also to assist with tasks that require human evaluation. The core contribution of our paper is transforming the evaluation task into an optimization problem over an evaluation tree, enhancing evaluation capability through preference branch selection.
>
> > Q: I think the writing needs improvement.
>
> 2. We will make revisions based on your suggestions:
>
> - Create a figure to illustrate how the evaluation task is performed. As described in section 3 (lines 91-95), the preference evaluation task involves taking two AI responses and determining which one better addresses the query (*win*, *lose*, and *tie*).
>
> - Include key conclusions in the captions. The main conclusions are already included in the paper, but if you have specific concerns, please let us know.
>
> > Q: just evaluating 5-6 models is far from enough. like model rankings?
>
> 3. New experiments. **We promise to include these experimental results in the next versions**:
>
> The quality and diversity of dialogue data are certainly crucial for training models. While collecting more diverse data is important, it is not the core contribution of our work. Our method can achieve significant performance improvements by incorporating more dialogue data, demonstrating its scalability and effectiveness.
>
> - We conduct Model Ranking, referring to [2] (all data from official github repo). Considering that our approach is more suited for Pair-wise Evaluation rather than Single Evaluation, the evaluation results might be biased. Nevertheless, our results exhibit a strong correlation with human scoring. *Experimental setup*: We used the DPO model under the OOD setting to score different models. We present our model's scores (Our Score) and the scores from the ChatBot Arena Leaderboard [2] as a reference (Arena Score). Delta represents the change in ranking.
>
> | |Arena Score|Our Score|delta|
> |--|--|--|--|
> |gpt-4 |1162|230.13|0|
> |gpt-3.5-turbo|1068|222.26| +2|
> |claude-v1|1149|219.53|-1|
> |wizardlm-13b |1058|218.93| +2 |
> |Llama-2-70b-chat|1093|217.80| -3|
> |vicuna-7b-v1.3|1005|212.46| +2|
> |Llama-2-13b-chat|1063| 210.86| -2|
> |Koala-13b|964|205.13|+1|
> |Chatglm-6b|924|200.16|+1|
> |RWKV-4-Raven-14B|921|197.66|+1|
> |Llama-2-7b-chat|1037|192.33| -4|
> |Alpaca-13B|901|183.06|0|
> |Dolly-V2-12B|822|165.13|0|
> |LlaMA-13B |799|160.93|0|
>
> (2) We conduct experiments with other baselines [3]. It is worth noting that our model was trained using only 7K dialogue data, significantly less than the 200K used by PROMETHEUS.
>
> |Model|Data|Eval-P|Eval-P|MT-bench|MT-bench|
> |--|--|--|--|--|--|
> |||w/ Tie|w/o Tie|w/ Tie|w/o Tie|
> |MIXTRAL-INSTRUCT-8X7B|| 53.81|73.50| 51.85| 71.42|
> |ULTRA RM (13B) | | |59.85|| 56.00|
> |PROMETHEUS-2-7B| 200k(label)|57.61|73.80| 56.18|67.25|
> |PROMETHEUS-2-8X7B| 200k(label)|58.41|79.98|55.07|71.96 |
> |**Ours(OOD DPO)**| 3k(label) + 4k(unlabel)|56.75|77.23|55.89|72.08|
>
> (3) We add experiments on the more general benchmark **Rewardbench** [4], which covers the AlpacaEval data you mentioned. **Please refer to the common response 5 for detailed experimental results.** These results effectively validate the efficacy of our method. Our model outperforms many commercial APIs and 70B parameter models. (Importantly, our experiments did not use any human labels, providing strong evidence for scalability.)
>
> The RewardBench evaluation dataset contains following:
> - Chat (alpacaeval-easy, alpacaeval-length, alpacaeval-hard, mt-bench-easy, mt-bench-medium)
> - Chat Hard (mt-bench-hard, llmbar-natural, llmbar-adver-neighbor, llmbar-adver-GPTInst, llmbar-adver-GPTOut, llmbar-adver-manual)
> - Safety (refusals-dangerous, refusals-offensive, xstest-should-refuse, xstest-should-respond, do not answer)
> - Reasoning (math-prm, hep-cpp, hep-go, hep-java, hep-js, hep-python, hep-rust)
>
> Our work demonstrates that (1) automated data construction can rival or even surpass human collected data, and (2) automated preference optimization methods can help evaluation models rapidly identify key evaluation criteria, ensuring efficiency in the evaluation process.
>
> > Q: Is it necessary to consider evaluations specifically for dialogue settings?
>
> 4. Thanks for your question. We will revise any inconsistent expressions in the paper accordingly (see common response 3). Furthermore, our experiments cover a broad range of domains, and we have supplemented experiments on Rewardbench, which should support the claims in our work and demonstrate the effectiveness of proposed method.
>
> [1] LLM Critics Help Catch LLM Bugs (OpenAI 2024)
>
> [2] Judging LLM-as-a-judge with MT-Bench and Chatbot Arena (NeurIPS 2023)
>
> [3] PROMETHEUS 2: An Open Source Language Model Specialized in Evaluating Other Language Models
>
> [4] RewardBench: Evaluating Reward Models for Language Modeling (Allen AI 2024)

---

> ### Comment · Reviewer_RFGW · 2024-08-12
>
> Thank authors for your responses. I keep my original score unchanged.

---

> > ### Comment · Area_Chair_gGHi · 2024-08-12
> >
> > Hi Reviewer RFGW, could you provide more insights on why / on which point you don't think the authors have sufficiently addressed your concerns (given that you decide to keep your score)? Because there is still one day until the end of the reviewer-author discussion period, they authors could still have chance to clarify and to provide more information to address your concerns.

---

> > > ### Comment · Reviewer_RFGW · 2024-08-13
> > >
> > > Hi AC,
> > >
> > > The authors have made a comprehensive rebuttal, which indeed clarified some concerns.
> > > However, I still feel many concerns cannot be simply solved by rebuttal–the first draft are only allowed to contain minor issues.
> > >
> > > I feel there are two main flaws of this paper:
> > > 1. Detached from the literature. Many works are lost in the introduction and related work section–discussion of the chatbot arena are missing; as a method for llms-as-a-judge evals, authors didn't explicitly discussed the recent llms-as-a-judge evals. This could reflects that this work could have been done without seeing sufficient research context.
> > > 2. The experiments demonstrating the effectiveness of their method are not convincing. The most important metrics, such as the correlation with chatbot arena requires at least 25 models to be statistically significant. And the original draft didn't contain such discussions, and the rebuttal also didn't show such correlation scores.

---

> > > > ### Author Response · Authors · 2024-08-13
> > > > **Further Discussion**
> > > >
> > > > Thanks for your review comments. Seems you have some misunderstandings about our paper.
> > > >
> > > > 1. **For concern 1**:
> > > >
> > > > + We have ensured that the majority of relevant works, particularly those related to preference evaluation with LLMs, have been cited [1-11].
> > > >
> > > > + The related works you mentioned on close-ended benchmarks are not focused on preference evaluation, while open-ended benchmarks, such as Chatbot Arena, are focus on model ranking. It’s important to note that model ranking is not the same as preference evaluation, even though it is conducted through a preference evaluation approach. Preference evaluation requires to evaluate diverse dialogue data, not just scoring different models (Because Chatbot Arena may not include some specialized user queries, please refer to [9]).
> > > >
> > > > + The Chatbot Arena and AlpacaEval that you referred to are part of the series of works cited as [1] and [11]. The only ones not cited are MMLU, GMS8k, and WildBench etc. We have already clarified in our common response that MMLU is used for accuracy evaluation. Our preference evaluation serves as a complement, primarily focused on evaluating human preferences.
> > > >
> > > > + We will include these missing citations in the next version.
> > > >
> > > > 2. **For concern 2**:
> > > >
> > > > + We do not consider *The correlation with Chatbot Arena* to be an indispensable metric. Please refer to recent works on preference evaluation [12] and [13], where most do not use Chatbot Arena correlation as a comparative experiment. Recent studies have centered their experiments around **Reward Bench**. We have added Reward Bench experiments in the common response, and even compared to the latest works, we maintain the novelty. Unlike [12], which used a large amount of human labeled data, and [13], which generated synthetic evaluation data using 70B LLMs without human labels, our approach consistently emphasizes "improving evaluation performance without using human labeled data." Additionally, we decomposed task difficulty using an evaluation tree, enabling a 7B model to comparable with a 70B model, which addresses a gap in [13] where small LLMs struggle with evaluation.
> > > >
> > > > + We have also provided the correlation coefficients between 20+ models and Chatbot Arena: Spearman correlation = 0.854.
> > > >
> > > > |                         | Arena Score | Our Score |
> > > > | :---------------------- | :---------- | :-------- |
> > > > | gpt-4                   | 1162        | 230.13    |
> > > > | mpt-30b-chat            | 1045        | 227.79    |
> > > > | vicuna-13b              | 1042        | 226.79    |
> > > > | gpt-3.5-turbo           | 1068        | 222.26    |
> > > > | zephyr                  | 1041        | 222.0     |
> > > > | claude-v1               | 1149        | 219.53    |
> > > > | wizardlm-13b            | 1058        | 218.93    |
> > > > | Llama-2-70b-chat        | 1093        | 217.80    |
> > > > | vicuna-7b-v1.3          | 1005        | 212.46    |
> > > > | Llama-2-13b-chat        | 1063        | 210.86    |
> > > > | Koala-13b               | 964         | 205.13    |
> > > > | oasst-sft-4-pythia-12b  | 893         | 203.5     |
> > > > | mpt-7b-chat             | 927         | 201.8     |
> > > > | Chatglm-6b              | 924         | 200.16    |
> > > > | RWKV-4-Raven-14B        | 921         | 197.66    |
> > > > | palm-2-chat-bison-001   | 1003        | 192.99    |
> > > > | Llama-2-7b-chat         | 1037        | 192.33    |
> > > > | Alpaca-13B              | 901         | 183.06    |
> > > > | stablelm-tuned-alpha-7b | 840         | 180.20    |
> > > > | Dolly-V2-12B            | 822         | 165.13    |
> > > > | LlaMA-13B               | 799         | 160.93    |
> > > > | fastchat-t5-3b          | 868         | 154.2     |
> > > >
> > > > [1] Judging llm-as-a-judge with mt-bench and chatbot arena
> > > >
> > > > [2] Gptscore: Evaluate as you desire
> > > >
> > > > [3] Tigerscore: Towards building explainable metric for all text generation tasks
> > > >
> > > > [4] Generative judge for evaluating alignment
> > > >
> > > > [5] Branch-solve-merge improves large language model evaluation and generation.
> > > >
> > > > [6] Self-critiquing models for assisting human evaluators
> > > >
> > > > [7] Large language models are not fair evaluators
> > > >
> > > > [8] Pandalm: An automatic evaluation benchmark for llm instruction tuning optimization
> > > >
> > > > [9] Evaluating large language models at evaluating instruction following
> > > >
> > > > [10] Judgelm: Fine-tuned large language models are scalable judges.
> > > >
> > > > [11] Alpacafarm: A simulation framework for methods that learn from human feedback
> > > >
> > > > [12] Foundational Autoraters: Taming Large Language Models for Better Automatic Evaluation （DeepMind 2024,7）
> > > >
> > > > [13] Self-Taught Evaluators （Meta 2024,8）

---

> > > > > ### Comment · Area_Chair_gGHi · 2024-08-14
> > > > >
> > > > > Hi Reviewer RFGW, do you think the authors' new response has addressed your concerns?

---

### Official Review · Reviewer_GhxJ · 2024-07-11

**Soundness:** 3
**Presentation:** 3
**Contribution:** 2
**Rating:** 4
**Confidence:** 3

**Summary:**

This paper proposes a novel approach to efficiently evaluate LLMs using branching preference learning. The authors conceptualize the evaluation process as a decision tree, where each path represents an evaluation reasoning trajectory. They introduce a tree-based data sampling method and preference learning based on the DPO algorithm to improve evaluation capabilities. The method is tested in three settings: in-distribution, out-of-distribution, and transfer evaluation. The authors claim their model significantly reduces dependency on labeled data and demonstrates strong performance across different evaluation settings while reducing inference costs by 90% compared to searching the entire evaluation tree.

**Strengths:**

- The paper's novel approach of framing LLM evaluation as a decision tree problem is a significant strength. This allows for a more nuanced and flexible evaluation process that can adapt to different scenarios and criteria. The use of branching preference learning enables the model to prioritize critical evaluation criteria.
- The authors test their model in multiple settings (in-distribution, out-of-distribution, and transfer evaluation), providing a thorough assessment of its performance. Applaud to that.

**Weaknesses:**

- The biggest concern I have is that in-distribution performance is not better than other baselines it compares to. This begs the question of where the improvement gain is from. If in-distribution evaluation performance is mediocre but out-of-distribution does better, then doesn't the most gain come from a better dataset?
- Another concern is the unnaturalness of using evaluation criteria as individual nodes. How to ensure the coverage of those criteria across different nodes. Are they overlapping each other or completely different? The paper is quite vague on this.
- Why is each criteria subtree only a binary tree? If using a tree structure, it seem like it can be easily extend to multiple nodes rather than just 2 at each layer.

**Questions:**

- It is unclear how to sample multiple evaluation path? By using a high temperature? Or is each individual criterion a criterion node?
- What is the Eval-P benchmark? It is not cited in the paper.

**Limitations:**

Yes

---

> ### Author Rebuttal · Authors · 2024-08-06
>
> Thank you very much for appreciating the novelty of our work. We hope the following responses can further address your concerns:
>
> > Q: If in-distribution evaluation performance is mediocre but out-of-distribution does better, then doesn't the most gain come from a better dataset?
>
> 1. We try to explain in detail why counter-intuitive results appeared in the in-distribution (ID) experiments, particularly as it may be related to "the curse of recursion" [1]:
>
> - First, there is a key difference between in-distribution (ID) settings and other (OOD and transfer) settings: the training data in the ID setting consists of high-quality, well-defined domain data collected by humans (including baseline AutoJ and Fennec). Therefore, in ID setting, the initial model, SFT model, and DPO model were all trained **using the same Dialogue data**. In other words, we iteratively synthesized new training data, and the queries for these data are the same. In contrast, the OOD and transfer settings **used different Dialogue data** (as described in lines 201-216 and in the Appendix).
>
> - Recent studies on iterative data synthesis have found that continuously using synthetic training data can led to model degeneration because the model tends to converge to a single modal distribution. As noted in [1], recursively using model-generated data can lead to "model collapse". This suggests that the training instability in our ID setting is likely caused by the use of synthetic data.
>
> - This phenomena align with our findings in the paper. In the OOD and transfer settings, where new training data is used, training remains stable. The new data typically constrains the model training process, helping it maintain a diverse distribution and preventing it from converging to a single distribution. Therefore, we recommend using dialogue data with different distributions at the SFT or DPO stages to prevent "the curse of recursion."
>
> - Despite this, the ID experiment results were still impressive, achieving the highest agreement score of 57.18 on Eval-P, a significant improvement over Fennec (note that in the ID experiment, our methods and Fennec used the same training data). All above evidence proves the effectiveness of our method.
>
> Regarding "most gain coming from a better dataset," the direct evidence is that the performance of our Initial model（trained on the our collected data）, is only 49.64, which is significantly lower than Fennec's 55.36. This shows that our data did not yield better results through direct training. However, considering another aspect of data or task diversity, our collected data is certainly better (see Sec 4.1 line 133-135). We sampled from a large-scale dialogue dataset rather than a specific data source. We then apply the K-Means algorithm to cluster the data. Subsequently, we sample data from these clusters, ensuring that the training dataset encompasses a diverse set of dialogue scenario. **This validates our motivation: even without human collected or labeled data, the model can achieve comparable or even better performance.** This also indicates that we can significantly reduce labor costs for preference evaluation tasks in the future.
>
> > Q: Are they overlapping each other or completely different?
>
> 2. In our experiments, only a minimal chance exists for the model to output semantically similar criteria. Our prompt is provided in Table 10, and Table 11 shows specific examples. It’s important to note that the idea of automatically generating criteria has been proposed in previous work [2]. Our contribution lies in guiding the model to prioritize criteria with high discriminative capabilities. Table 1 shows that we can achieve better results with fewer inference steps (branches).
>
> > Q: Why is each criteria subtree only a binary tree?
>
> 3. You are correct that evaluation trees can extend to multiple nodes. However, considering: (1) **Computational efficiency**, each sample to be evaluated includes 10 criteria (k=10), each criterion includes 2 scoring guidelines, followed by 4 different judgments (see lines 151-157). Thus, there are 80 different subtrees. The purpose of building the evaluation tree is to expand the search space, and the **current method already has a high computational complexity, making additional children unnecessary.** (2) Scoring guidelines and judgments are sampled by adjusting temperature and swapping response positions. The diversity obtained by adjusting temperature is limited and requires different conditions (criteria and scoring guidelines). (3) **Another purpose of constructing the evaluation tree is to obtain preference pairs (chosen, reject) for DPO training**. Clearly, a binary tree is sufficient.
>
> > Q : It is unclear how to sample multiple evaluation path? By using a high temperature?
>
> 4. In lines 151-154, we have stated how different tree nodes are generated, i.e., criteria are ensured to be diverse by adjusting the model’s prompt, while scoring guidelines and judgments are obtained by adjusting temperature (specifically, judgments can be obtained by swapping response positions).
>
> > Q: What is the Eval-P benchmark?
>
> 5. Eval-P was proposed by AutoJ [3]. We will add the citation at line 199.
>
> We also provide a more detailed background and related work to clarify our work, as well as clearer contributions and new experimental results. Please refer to the common response (Author Rebuttal). We hope this can further address your concerns.
>
> [1] AI models collapse when trained on recursively generated data（Nature）
>
> [2] Branch-solve-merge improves large language model evaluation and generation （Meta 2023）
>
> [3] Generative judge for evaluating alignment (ICLR 2024)

---

> > ### Comment · Reviewer_GhxJ · 2024-08-12
> >
> > Thank you for the response and addressing my questions. I am still concerned about the ID setting shown in Table 1. On the same dataset, compared to Auto-J and Fennec the proposed method is not really doing better. It is sometimes worse. This is evidence that the methodology is not the reason behind performance gain. Thank you for bringing up model collapse. One can definitely avoid model collapse even during iterative data synthesis [1]. The main problem here is that external data seems to be helping the most rather than the evaluation tree process.
> >
> > [1] https://x.com/RylanSchaeffer/status/1816535790534701304

---

> ### Author Response · Authors · 2024-08-14
> **Further Discussion**
>
> **Thank you very much for your suggestions, which have helped us improve the quality of our work.** Following your advice, we have conducted new experiments and  to provide the following clarifications：
>
> + We have never denied the contribution of data, especially the additional unlabeled data (in the OOD setting) used in our approach. However, it is important to note that this data is unlabeled, making it widely accessible without increasing extra costs.
>
> + These additional data are simply dialogue data. The purpose of the evaluation tree is to generate the criteria, scoring guidelines, and judgments required to train the evaluation model. Therefore, including only dialogue data is insufficient for training an evaluation model. Hence, we cannot consider the new unlabeled dialogue data as the key to improving performance. Our approach requires combining these dialogue data and constructing SFT and DPO data.
>
> + **Most importantly**: Our proposed method is based on the observation in Figure 1, which shows that increasing the number of branches can enhance evaluation performance. This means that even without the additional data, the initial model can achieve good evaluation performance (**though it may require > 40 branches**). The SFT and DPO methods are designed to enable the model to efficiently reach the final result (**< 3 branches**). This is also why we employed branch ensemble in Section 4.4, as branch ensembles generally yield better decision results and provide a higher upper bound on performance. The purpose of designing the evaluation tree is to allow the model to more quickly identify the key evaluation paths, which can be derived from the branch ensemble results (as referenced in Section 4.4). **Thus, the evaluation tree is largely aimed at accelerating the evaluation process by sampling fewer branches (yet more crucial)**. I believe our method's design and experimental results are well-supported by the substantial evidence mentioned above.
>
> + We verified whether model collapse was the reason for the impact on ID performance. By mixing the original training data with the newly synthesized data when training the SFT model according to your advise (thanks again), **we observed significant performance improvement, which supports our initial hypothesis in Section 5.7**. Continuously training the SFT model and DPO model with the same data affects model convergence. However, compared to the OOD setting, there is still a lack of new unlabeled data, making direct comparisons still unfair. If we obtain more unlabeled domain-specific data, we believe our methods can achieve even better results. Considering the rebuttal time limitation, we may not be able to conduct additional experiments. However, our experiments, along with the common response experiments on the Reward Bench, should be sufficient to address your concerns.
>
>
> |                                    | branch | Eval-P w/tie    | Eval-p w/o Tie  |
> | ---------------------------------- | ------ | --------------- | --------------- |
> | Ours ID SFT (copy from paper)      | 1      | 56.68/86.64     | 70.76/89.11     |
> |                                    | 5      | 55.96/86.57     | 72.91/88.13     |
> | Ours ID DPO (copy from paper)      | 1      | 55.24/84.26     | 69.87/86.95     |
> |                                    | 5      | 57.18/85.63     | 74.88/88.52     |
> | Ours OOD SFT (copy from paper)     | 1      | 54.59/87.14     | 70.56/88.52     |
> |                                    | 5      | 55.10/87.86     | 73.69/89.99     |
> | Ours OOD DPO (copy from paper)     | 1      | 55.89/89.44     | 75.76/90.67     |
> |                                    | 5      | 56.75/90.37     | 77.23/92.24     |
> | Ours ID SFT + Mix Trainng  （new） | 1      | 56.87/87.92     | 70.92/88.99     |
> |                                    | 5      | 56.32/87.86     | 75.17/89.79     |
> | Ours ID DPO + Mix Trainng  （new） | 1      | 57.54/90.66     | 78.60/93.32     |
> |                                    | 5      | **58.41/91.52** | **79.69/93.72** |
>
> Caption：Our Mix Training setup differs from that in the paper by using both original and synthetic data to train the model, whereas the paper only used synthetic data for training the sFT model. As a result, we achieved an Agreement of 58.41 vs. 55.80 , and a Consistency of 79.69 vs. 74.14 compared to Fennec.

---

> > ### Comment · Area_Chair_gGHi · 2024-08-14
> >
> > Hi Reviewer GhxJ, do you think the authors' new response has addressed your concerns?

---

### Official Review · Reviewer_cKwh · 2024-07-12

**Soundness:** 2
**Presentation:** 2
**Contribution:** 2
**Rating:** 4
**Confidence:** 2

**Summary:**

They present an approach to improving LM evaluation by having models first generate an evaluation criteria, then a scoring guideline, and then finally a final judgement. They then develop a procedure for collecting training data corresponding to these three steps by applying branching/pruning approach (sample multiple criteria, from each sample multiple guidelines, etc...). They then use the generated data to train a DPO and SFT model. They find that their method outperforms baseline evaluation approaches according to correlation with human judgement on dialogue evaluation.

**Strengths:**

* The problem of improving LM evaluation is important
* The idea of enabling language models to hierarchically sample evaluations (e.g. fist criteria, then guideline, then judgement) is very neat, and similarly the idea of applying a tree-based sampling procedure to automatically generate data is quite cool.
* I think they do fairly thorough experiments and compare to quite a few baselines.

**Weaknesses:**

* The paper is honestly pretty hard to follow. There's a lot of moving parts and it's not explained in an easy to digest way.
* The specific method presented seems a little bit ad-hoc, and could be justified better in the paper (e.g. why use criteria, then guideline, then judgement, why not some other sequence of steps?).
* Looking at Figure 1, it doesn't seem that their method improves all that much over the baseline

Nits:
* The related work seems pretty sparse. There's lots of work on improving LM evaluation in math reasoning settings that isn't discussed.
* Figure 4, the text is really small and hard to read.

**Questions:**

* The description of the growth / pruning part is pretty confusing and in general a better explanation of what exactly is happening there would be really helpful.
* "Agreement quantifies the proportion of evaluations that meet the criteria for swap consistency and align with human judgments.": why both of these criteria need to be met? Isn't the swap part already covered by the consistency evaluation?

**Limitations:**

They do a good job of discussing the limitations. I would also note that it is unclear how effective this is when applied to more challenging tasks like mathematical reasoning (e.g. MATH benchmark) as a limitation.

---

> ### Author Rebuttal · Authors · 2024-08-06
>
> Thank you very much for appreciating our idea of transforming the evaluation task into a tree search problem. We hope the following responses can further address your concerns：
>
> > Q: The paper is honestly pretty hard to follow. There's a lot of moving parts and it's not explained in an easy to digest way
>
> 1. Preference evaluation is indeed a rapidly developing field. We have detailed the task background, related work, and our contributions in the common response, hoping to alleviate your concerns. Additionally, considering that Reviewer YdUv Strengths 2 and Reviewer sKgh Strengths 1 acknowledge our work clear and easy to understand, could you please specify which parts is difficult to comprehend? This will help us make better revisions.
>
> > Q: The specific method presented seems a little bit ad-hoc, and could be justified better in the paper
>
> 2. We believe our method for decomposing the evaluation task is reasonable, though not the only possible approach. Using a different decomposition method **would not diminish** our contributions: (1) Human evaluations also rely on criteria, scoring guidelines, and judgments. Recent research [1] has highlighted the need for multiple rounds of negotiating these criteria and guidelines for human evaluators. Another study [2] employed an automated evaluation process. There are certainly better ways to decompose evaluation tasks, and we are open to exploring them further. (2) Our primary focus is not on the decomposition method itself, but on how to avoid relying on human supervision and efficiently identify the key evaluation criterion, as discussed in common response 4.
>
> > Q: Looking at Figure 1, it doesn't seem that their method improves all that much over the baseline
>
> 3. (1) Our method shows significant performance improvement over the initial model (w/ tie 55.89 vs. 49.64 w/o tie 75.76 vs. 57.02). (2) For the AutoJ and Fennec methods, it's important to note that our settings differ. Our experiments are conducted in OOD scenarios, using training sets samples from large-scale data rather than human collected (Figure 4 illustrates the differences in data domains). **Additionally, we have added new experiments in the common response 5** to demonstrate the effectiveness of our method. On the RewardBench, our model outperforms the 70B LLaMA3 and the Prometheus v2, which was trained on 200K  evaluation data, while our training data accounts for only 10% of theirs.
>
> > Q:  The related work seems pretty sparse.
>
> 4. As discussed in the common responses 1 and 2, we will add more related work to address your concerns. It's important to clarify that the challenges in preference evaluation (LLM evaluation) include not only accuracy evaluations like MMLU and Math Reasoning but also **open-ended generation** that requires multiple evaluation dimensions. Our work aims to address multi-dimensional evaluation in complex and diverse user intent scenarios, as mentioned in the abstract (line 4: Particularly, in complex dialogue scenarios involving diverse and intricate user intents). Furthermore, our experiments include *Code* and *Math* test data in Table 2, reflecting improvements in these areas. **Our new experiments on RewardBench show significant performance improvements in reasoning tasks (nearly 10 points, as referenced in the common response)**.
>
> > Q: Figure 4, the text is really small and hard to read.
>
> 5. We will add a Table for Figure 4 to explain the percentage of each category. Figure 4 illustrates the color differences in various domains, indicating significant differences in the constructed OOD data domains, which are more challenging.
>
> > Q:  The description of the growth / pruning part is pretty confusing and in general a better explanation of what exactly is happening there would be really helpful.
>
> 6. I'm not sure what caused your confusion. If you could specify, it would help us make better revisions. In fact, we have attempted a clearer explanation: The purpose of constructing an evaluation tree: (1) to expand the candidate space of the evaluation process and (2) to create SFT data for different evaluation paths and (chosen, reject) DPO data. Our method achieves goal (1) through heuristic sampling and temperature sampling (lines 151-154). Goal (2) is accomplished using two consistency pruning methods, with consistent samples used as SFT data and inconsistent ones as DPO data (lines 158-161).
>
> > Q:  why both of these criteria need to be met? Isn't the swap part already covered by the consistency evaluation?
>
> 7. Agreement (AGR) reflects the consistency between LLM evaluations and human judgments. Consistency (CNS) represents the extent of bias introduced after swapping positions, indicating the stability of the model's evaluations. Please refer to [2].
>
> [1] A Holistic Approach to Undesired Content Detection in the Real World (OpenAI 2023)
>
> [2] Branch-solve-merge improves large language model evaluation and generation （Meta 2023）
>
> [3] Generative judge for evaluating alignment (ICLR 2024)

---

> > ### Comment · Reviewer_cKwh · 2024-08-13
> >
> > I appreciate your response. If you could add some more clarity in the paper about how the AutoJ and Fennec settings differ from yours, that would be great.
> >
> > Regarding the confusing parts: I think it would be less confusing with a clear example or an LM evaluation using your framework somewhere in the main text (e.g. a concrete example in a figure somewhere). Your method has a large description length, which generally goes against my prior, and makes more skeptical that every aspect of the method is necessary and also muddles the true contribution of the work. I think positional consistency is a pretty random heuristic. I agree that LMs might be sensitive to position and whatnot, but LMs are sensitive in all kinds of ways to prompts, so why focus on just position; you could invent so many other arbitrary heuristics that would probably be just as good and provide no real insight whatsoever. Pair these criticisms with the fact that one of the evaluation metrics is less standard (e.g. consistency) (essentially created to show that this paper's method works), I would have a hard time accepting the paper as it stands. That being said I think the reward bench evaluation you added looks promising, so I am willing to raise by 1 point.

---

> ### Author Response · Authors · 2024-08-14
> **Further Discussion**
>
> Thanks for your review comments!
>
> > If you could add some more clarity in the paper about how the AutoJ and Fennec settings differ from yours:
>
> We outline the settings and core challenges, highlighting the differences:
>
> + Auto J's approach evaluates two different AI responses using a one-step evaluation process. The dialogue data in its training dataset was human created to match the distribution of the test set, as shown in Figure 4. During training, only the SFT method was used, with no DPO training included.
>
> + Fennec employs a multi-step evaluation, using criteria and scoring guidelines to alleviate the complexity of the evaluation task. The training also involved human selected dialogue data and included only SFT training, with no DPO training.
>
> + In our approach: 1）There is no human selection or labeling cost—both the sampling of dialogue data and preference labels are fully automated. 2）We use a multi-step evaluation process to simplify the difficulty of complex evaluation tasks. 3）Most importantly, we incorporate DPO optimization, which helps the model achieve better performance with fewer branches, enhancing evaluation effectiveness.
>
> + Our method demonstrates that even a 7B parameter evaluation model has significant potential compared to the 70B model used in [1] for evaluation tasks. This advantage is largely due to our task decomposition (similar to Chain of Thought), which breaks down the evaluation process into three steps: generating evaluation criteria, scoring guidelines, and final judgements results. These steps and their intermediate outputs simplify the task, making the 7B model more capable in complex scenarios.
>
> > Your method has a large description length, which generally goes against my prior, and makes more skeptical that every aspect of the method is necessary and also muddles the true contribution of the work.
>
> + If possible, could you please point out the details that might be unclear to you? This will help us explain our work more clearly.
>
> + We will also include a new Figure to illustrate how the evaluation task is conducted: The evaluation model receives a user query and two AI responses, and it only needs to assign one of three labels (win, lose, or tie) to indicate which AI response is better, or if they are equal.
>
> + In the Methods section (Section 4), we cover the following: 1) Section 4.1 explains how we collect the dialogue dataset. 2) Section 4.2 shows how we train an initial model. 3) Section 4.3 details how we construct the evaluation tree. 4)Section 4.4 describes how we collect training data based on the evaluation tree. 5) Section 4.5 demonstrates how we train the SFT and DPO models. Among these, only Section 4.2 follows a process similar to other related work. The remaining sections introduce new approaches developed in our work, which may explain some discrepancies with your prior understanding. Even compared to the latest research [1], our work offers novel contributions: We demonstrate that it's possible to improve evaluation model performance without relying on human labeled data, and we also showcase the potential of a 7B model, which is popular in both academia and industry.
>
> > I think positional consistency is a pretty random heuristic. I agree that LMs might be sensitive to position and whatnot, but LMs are sensitive in all kinds of ways to prompts, so why focus on just position
>
> + The issue of positional consistency was not proposed by us; please refer to references [2] and [3].
>
> + Given that works [3], [4], and [5] all consider positional consistency and use Agreement and Consistency as evaluation metrics, I do not agree this as a flaw in our work.
>
> + We fully agree with your point that LLMs exhibit more bias beyond positional consistency. However, since positional consistency is the most direct and easily improvable, I believe it is necessary to address these biases during training. In fact, with pirwise evaluation, if positional consistency significantly decreases when positions are swapped, it would be unacceptable for downstream applications such as reward models or model corrections.
>
> Considering that both Reviewer YdUv and Reviewer sKgh also agree our presentation to be clear, we hope to engage in further discussion to address any concerns for you.
>
> [1] Self-Taught Evaluators （Meta 2024,8）
>
> [2] Large language models are not fair evaluators
>
> [3] Generative judge for evaluating alignment
>
> [4] PROMETHEUS 2: An Open Source Language Model Specialized in Evaluating Other Language Models
>
> [5] OffsetBias: Leveraging Debiased Data for Tuning Evaluators

---

> > ### Comment · Area_Chair_gGHi · 2024-08-14
> >
> > Hi Reviewer cKwh, do you think the authors' new response has addressed your concerns?

---

### Official Review · Reviewer_sKgh · 2024-07-17

**Soundness:** 4
**Presentation:** 4
**Contribution:** 3
**Rating:** 6
**Confidence:** 3

**Summary:**

The paper investigates how to improve the quality of automated evaluation through fine-tuning (SFT and DPO). The main algorithm proposed by the paper is to construct an search tree which consists of node of (criterion, scoring guide, and judgment). This tree is later pruned and modified and the different paths serve as fine-tuning data for SFT and DPO.

My current rating is tentative. If the authors can kindly clarify the details of the paper, I'm happy to raise the score.

**Strengths:**

1. The paper is very clear and easy to read.
2. The investigation is very thorough. Experiment is comprehensive (the in-distribution, out-of-distribution evaluation setup is great).
3. The main claim of the paper is substantiated (I.e., improving efficiency through fine-tuning).

**Weaknesses:**

I don't think this paper has substantial weaknesses.

1. There are some imperfections of text -- mostly just need to be clarified. Missing notation definitions, etc.
2. The performance improve over Auto-J on AGR is minor (55.13 -> 57.18). OOD evaluation, Zephyr-7B AGR is 56.75 and GPT-4 is 62.28 (which is close, but not quite close). CNS however is beating GPT-4. This would be very helpful for me to understand a bit more about what CNS is, and whether beating GPT-4 on this metric is meaningful or not (see Q6).

**Questions:**

I do however have a few questions:

1. It seems that no human preference/judgment label is used. The initial policy is GPT-4. Then the creation of data for SFT and DPO are through the tree construction. Is my understanding correct? If so, this is a type of self-improving/bootstrapping method, which I find quite interesting. It also has greater impact on other fields of LLM.
2. Sec 4.2, consistency pruning. I'm not sure I understand self-consistency -- In Figure 3 left most tree, how did you obtain different versions of J1? The figure label says "red dot is judgment result by swapping the response positions" -- isn't it "positional-consistency" if swapping positions is involved? Please clarify.
3. When you collect preference label (Sec 4.4), for Branch Ensemble, how do you create the ensemble?
4. LLM-as-a-Judge. The writing says additional analysis is in Section 5.4. Sec 5.4 talks more about Auto-J performance in different scenarios. Where is the result of using Auto-J as a judge to create labels for DPO?
5. Table 2 Auto-J dagger (second row). What is this!? The caption did not explain what this means. It might be in the text but I couldn't find it.
6. Can you expand on why AGR and CNS are good evaluation metrics? Appendix A.2 only has training details.
7. I appreciate the honesty in Sec 5.7 -- can you offer some explanation on why DPO is unstable for in-distribution training? It's often hard to explain empirical behaviors -- it's perfectly ok if you don't have an explanation.

**Limitations:**

Yes

---

> ### Author Rebuttal · Authors · 2024-08-06
>
> Thank you very much for your appreciation of our work. We hope the following responses can address your concerns:
>
> > Q: It seems that no human preference/judgment label is used.
>
> 1. You are right that we do not use any human labels (see common response 4). As you can see, many research fields and studies lack sufficient resources (time and money) to employ annotators for labeling preference data. However, the APIs from OpenAI or Claude are also very expensive, and we aim to address these problems by providing open-source evaluation datasets and models to contribute to the community.
>
> > Q: how did you obtain different versions of J1?
>
>
> 2. Yes, there are two types of self-consistency here: (1) Consistency after swapping positions (yellow dot J1 and red dot J1) and (2) Consistency due to sampling temperature (tmp>0) leading to different judgments (J1 and J2). Thus, we obtain four evaluation results, and these judgments should be consistent, based on the same criteria and scoring guidelines.
>
> > Q: how do you create the ensemble?
>
> 3. The idea of using a model ensemble stems from our findings in Figure 1, which show that increasing the number of branches significantly improves the model's inference performance. Therefore, we increase the number of branches during inference (to about 80) to collect the model's preference labels (including *'win', 'lose', 'tie'*). We then use the ensemble results from all branches as the final preference labels.（It is worth noting that the ensemble is highly effective because the classification labels are limited, although we still approach this as a generative task rather than a classification task.）
>
> > Q: Where is the result of using Auto-J as a judge to create labels for DPO?
>
> 4. The results in "w/ GPT-4" are those evaluated using GPT-4 for preference evaluation (rather than dialogue evaluation) and used for training. However, the improvement may not be significant as shown in Sec 5.4. We suspect this is because the preference evaluation task is also challenging for GPT-4, further demonstrate the importance of this area we are exploring.
>
> > Q: Auto-J dagger (second row). What is this!?
>
> 5. In line 218, we explain the meaning of the dagger (it represents our reproduced results). We will also clarify this in Table 2 to ensure a clearer description.
>
> > Q: Can you expand on why AGR and CNS are good evaluation metrics?
>
> 6. The purpose of preference evaluation is to measure alignment with human preferences, evaluating whether different AI responses better reflect human behavior or values. Agreement (AGR) [1] reflects the consistency between LLM evaluations and human judgments, serving as a direct metrics. Consistency (CNS) represents the extent of bias introduced after swapping positions, indicating the stability of the model's evaluations. Currently, it is a relatively good until we find more precise and fine-grained metrics.
>
> - To address your observation on *"The performance improve over AGR is minor, compared to GPT-4"*:
> Firstly, it's important to note that **most responses can be judged as either "win" or "lose," while a "tie" typically indicates that an effective discriminative criterion has not been identified (even with human evaluations).**
> If a more suitable criterion or strict scoring guidelines were applied, many of these "tie" labels could be classified as either "win" or "lose."
> From this perspective, our method effectively identifies more discriminative criteria and scoring guidelines. This allows cases that human evaluators might label as "tie" to be classified as either "win" or "lose." Consequently, this leads to a more noticeable AGR improvement in scenarios "w/o tie" labels. This directly supports our claims and demonstrates the effectiveness of our method.
>
> - *"why CNS however is beating GPT-4?"*
> Our evaluation models mitigate position bias by training on data where response positions are swapped. In our DPO training, the consistency of swapping positions of resposne is included in our training objectives and datasets (line 158-161).
>
> > Q: can you offer some explanation on why DPO is unstable for in-distribution training?
>
> 7. We are very happy to discuss this issue, particularly as it may be related to the "curse of recursion" [2]:
>
> - First, there is a key difference between in-distribution (ID) settings and other (OOD and transfer) settings: the training data in the ID setting consists of high-quality, well-defined domain data collected by humans. Thus, in our experiments, both the SFT and DPO processes use the same dialogue data. In other words, we iteratively synthesized new training data, and the queries for these data are the same. In contrast, new data is used in the OOD and transfer settings (as described in the Experiments and Appendix).
>
> - Recent studies on iterative data synthesis [2] have found that continuously using synthetic training data can led to model degeneration because the model tends to converge to a single modal distribution.
>
> - This phenomena align with our findings in the paper. In the OOD and transfer settings, where new training data is used, training remains stable. The new data typically constrains the model training process, helping it maintain a diverse distribution and preventing it from converging to a single distribution.
>
> - Therefore, we recommend using dialogue data with different distributions at the SFT or DPO stages to prevent the "curse of recursion."
>
> We also provide a more detailed background and related work to clarify our work, as well as clearer contributions and new experimental results. Please refer to the common response (Author Rebuttal). We hope this can further address your concerns.
>
> [1] Generative judge for evaluating alignment (ICLR 2024)
>
> [2] AI models collapse when trained on recursively generated data（Nature）

---

> > ### Comment · Area_Chair_gGHi · 2024-08-14
> >
> > Hi Reviewer sKgh, do you have any comments regarding the authors' rebuttal?

---

### Official Review · Reviewer_YdUv · 2024-07-23

**Soundness:** 3
**Presentation:** 3
**Contribution:** 3
**Rating:** 6
**Confidence:** 2

**Summary:**

In this work, the authors propose a tree-based data sampling method to conceptualize the evaluation process as a decision tree, where each node represents an evaluation action, and each path from the root to a leaf node represents a trajectory of evaluation reasoning. The proposed method involves generating supervised data and preference pairs derived from the evaluation tree for SFT and DPO training. This approach aims to reduce the dependency on labeled data and improve the performance of the evaluation model across in-distribution, out-of-distribution, and transfer evaluation settings. Experimental results demonstrate that the proposed model can enhance evaluation efficiency and performance.

**Strengths:**

1. The proposed method reduces the dependency on human labeled data by generating supervised data and preference pairs from the evaluation tree.
2. The paper is well-written.

**Weaknesses:**

1. Potential Biases --- The initial multi-branch training data is generated using only GPT4, which could introduce bias to the training data. Moreover, the branch ensemble method could also introduce bias to the training data. If the training data is biased or unrepresentative, the model's evaluations may also be biased. The authors should consider labeling a small annotation set to validate the branch ensemble approach.

**Questions:**

1. In table 1, what is the number of branches generated by Initial, SFT, and DPO for evaluation?

Typos:
1. Line 36: single quote format error
2. Line 256: "" -> ``''
3. Figure 4: font too small, hard to read

**Limitations:**

Yes, limitation discussed in the paper.

---

> ### Author Rebuttal · Authors · 2024-08-06
>
> Thanks for your review comments, as well as your appreciation of the importance and writing quality of our work.
>
> **For Concerns：**
>
> > Q: Potential Biases
>
> We believe that incorporating synthetic data is essential for the future development of LLMs. Of course, reducing bias is also a crucial issue (points 1 and 2), and while we can attempt to mitigate the harm caused by biases, we cannot completely eliminate them (point 3). In our experiments and settings, the impact of bias is relatively small because our evaluation align with human evaluation results (point 4).
>
> 1. The use of synthetic data has become indispensable in both research and industry, enhancing performance [1, 2, 3] and supporting theoretical analysis [4]. OpenAI's GPT-4, one of the most widely used models today, excels in both safety and bias, having undergone extensive user testing with no more comparable open or closed-source alternatives.
>
> 2. One of our key contributions is exploring how to use LLMs to make improvements **without rely on human supervision (line 39)**. Based on this condition, we did not introduce additional high-quality human-supervised data. Of course, we believe that incorporating such data could further enhance model performance and reduce bias。
>
> 3. In fact, both humans and LLMs inherently exhibit bias (though not all biases affect practical use). Addressing how to mitigate bias should be part of a comprehensive discussion, and we will further explore the impact of data bias on evaluation in future work. "Labeling a small annotation set to validate the branch ensemble approach" cannot be effectively done until we clearly understand the types of biases introduced.
>
> 4. Given that our experimental metric is "agreement and consistency with human preferences," our results demonstrate that our method achieves judgments more aligned with human evaluations (as our experimental evaluation are based on human evaluation results). This indicates that our approach does not introduce significant bias. If any bias exists, it likely aligns more closely with human preferences rather than originating from synthetic data.
>
> **For Questions:**
>
> > Q: what is the number of branches generated by Initial, SFT, and DPO for evaluation?
> 1. The numbers in the third column of Table 1 (Branch) indicate the number of branches used in inference or evaluation. （For obtaining training data, we sample 10 criteria, each with 2 scoring guidelines, and each score includes 4 judgments, resulting in a total of 80 = 10 * 2 * 4 branches.）
>
> We have also outlined the background and significance of current task, along with our contributions, in the common response (Author Rebuttal), which we hope addresses your other concerns.
>
> [1] LLM Critics Help Catch LLM Bugs (OpenAI 2024)
>
> [2] The Llama 3 Herd of Models (Meta 2024)
>
> [3] Phi-3 Technical Report: A Highly Capable Language Model Locally on Your Phone (MicroSoft 2024)
>
> [4] Physics of Language Models: Part 3.3, Knowledge Capacity Scaling Laws (ICML 2024)

---

> > ### Comment · Reviewer_YdUv · 2024-08-11
> >
> > Thanks to the authors for the responses. I will keep the current scores that I believe reflects my overall assessment of the paper.

---

### Author Rebuttal · Authors · 2024-08-06

We are grateful to Reviewer cKwh, GhxJ, and RFGW for appreciating the novelty and interest of our approach. We also appreciate the acknowledgment of our writing and presentation by Reviewer YdUv and sKgh.

We need to make the following clarifications: how our research differs from other work (1 and 2); the specific research area our work focuses on (3); the contributions of our work (4); and the new experimental results (5).

1. First, I need to clarify that "preference evaluation" have become one of the most rapidly developing research areas since the advent of LLMs. As noted in the review, there is still relatively little related work in this emerging field.  Unlike previous evaluation benchmarks such as MMLU, GSM8K, and many other automated evaluation tasks that contain golden answers to evaluate accuracy, "preference evaluation tasks" have the following characteristics:
+ These tasks **do not have golden answers**, such as open-domain dialogue or summarization tasks, which largely rely on human evaluation. For instance, in [1], human evaluation scores improved, but automated metrics did not.
+ These tasks **have multiple evaluation dimensions (criteria)** and cannot rely solely on accuracy metrics. For example, in coding and math problems, the *reasoning steps*, *logical*, and *readability of the format* are often more important than just getting the correct answer, as reported in [2].
+ The primary importance of these tasks is their alignment with human preferences, which is a crucial aspect of current alignment training. Therefore, the methods, such as PandaLM [3], AutoJ [4], Promthus [5], and BSM [6], aim to enhance consistency with human preferences, providing solid baselines and related work for our research.

Additionally, even in scenarios where correctness is the primary metric, we often need to rank AI responses that are both incorrect, determining which one is better (i.e., requiring a more fine-grained score rather than simply correct or incorrect). Therefore, preference evaluation serves as a valuable supplement to the existing benchmarks.

2. We have indeed discussed and defined "preference evaluation" (lines 3-5) and their distinction from other evaluation methods (lines 23-32) in our work. Additionally, we have talk about the differences between automated metric (ROUGE) and LLM evaluations. As previously mentioned, benchmarks like MMLU are solely automated metric evaluations and do not encompass multi-dimensional, human-aligned preference evaluations. To provide a clearer and more comprehensive distinction of this task, **we promise to include more relevant references (such as MMLU, MMLU-pro, MixEval) in the introduction and related work sections**.

3. In our paper, some terms like "Dialogue evaluation" and "LLM evaluation" might cause confusion. We are actually focus on "preference evaluation using LLMs in open-domain dialogue tasks." we will keep consistent description in next version.

4. Our goal is to verify that **if we can enhance LLM evaluation capabilities without introducing human-supervised signals？**. The contributions are as following:
+ Our methods and experiments support this idea: (1) training data distributions are automatically constructed (methods 4.1). (2) the SFT and DPO training data (preference labels) are synthesized by LLMs (methods 4.3 and 4.4), with the only external signal being the initial model data generated by GPT-4. Thus, compared to the scenario in Figure 1, our exploration is more realistic and challenging.
+ We are the first to model and optimize "how to find appropriate evaluation criteria," significantly reducing the number of branches. This is supported by our observation that **increasing the number of model branches (up to 40 branches) yields better judgment results** (Figure 1 and lines 49-54). Our method optimizes the criteria space, allowing the model to achieve high performance with evaluations involving just 1-3 branches.
+  Scoring preference data for preference evaluation is currently challenging (this refers to evaluation preferences, not just preferences between two dialogues). We discovered that **branch ensemble can provide a better judgment upper limit (though not a golden label)**. Using this approach, we are also the first to introduce evaluation tree and DPO training in the evaluation model's training (to our knowledge).

5. We further validated the effectiveness of our method on RewardBench  [8], which is very popular in the community for verifying preference alignment. We collected additional 10K dialogues, resulting in a total of 16K SFT data from the paper and 6K unlabeled data for DPO training. Despite this, it is still a small compared to the 200K training data of Prometheus v2 [7]. We observed that our methods significantly improves model performance (76.1 vs. 72.1), especially on reasoning tasks, with nearly a 10 point increase.

| |Score|Chat|Chat Hard|Safety|Reasoning|
|--|--|--|--|--|--|
|GPT4| 85.9|95.3|74.3| 87.2 | 86.9|
|**Ours (DPO)** |76.1| 95.7|51.4|83.5| 73.6|
| Llama3-70B-Instruct|76.0| 97.6 |58.9| 9.2|78.5|
| Calude3-sonnet-0229|75.7|93.4|56.6 |83.7|69.1|
| Prometheus-8X7B-v2.0 |75.3|93.0|47.1|83.5|77.4 |
| Prometheus-7B-v2.0|72.4| 85.5| 49.1| 78.7| 76.5|
| **Ours (SFT)** |72.1| 93.6 | 51.6| 82.7|60.3 |
| Llama3-8B-Instruct | 64.8| 85.5| 41.6|67.5|64.8|

**reference:**

1 Learning to summarize from human feedback (Neurips 2020)

2 PROVER-VERIFIER GAMES IMPROVE LEGIBILITY OF LLM OUTPUTS (OpenAI)

3 Pandalm: An automatic evaluation benchmark for llm instruction tuning optimization (ICLR2024)

4 Generative judge for evaluating alignment (ICLR2024)

5 Prometheus: Inducing fine-grained evaluation capability in language models (ICLR2024)

6 Branch-solve-merge improves large language model evaluation and generation （Meta2023）

7 PROMETHEUS 2: An Open Source Language Model Specialized in Evaluating Other Language Models

8 RewardBench: Evaluating Reward Models for Language Modeling (Allen AI 2024)

---

### Decision · Program_Chairs · 2024-09-25

**Decision:**

Reject

**Comment:**

In this paper, the authors conceptualize the NLG evaluation process as a decision tree, where nodes are evaluation actions (criteria, scoring guidelines, or judgements) and paths are evaluation trajectories consists of a sequence of actions. To fine-tune an LM to generate NLG evaluation following such procedure, the authors propose to collect data by sampling branches from such trees while applying some pruning heuristics (self-consistency and positional-consistency). The authors show that LMs fine-tuned with such data (with SFT and DPO) outperform baseline systems on benchmarks such as Chatbot Arena and RewardBench.

All reviewers agree that the idea of framing LLM evaluation as a decision tree is novel and interesting; they acknowledge that reducing the dependency on human-labeled data by generating supervised data and preference pairs from such evaluation tree could be empirically very useful; most reviewers agree that the authors did a good job on their experiment design, they have included multiple settings (in-distribution, out-of-distribution, and transfer evaluation) and provided a thorough assessment of the proposed method against many baselines systems.

Reviewers have mixed opinions regarding the presentation of the work, YdUv and sKgh find it easy to read but cKwh and RFGW find the method description less straightforward and suggest the authors to provide a figure demonstrating concretely how an LM evaluation is conducted using the proposed framework.

During the author-reviewer discussion period, the authors have successfully addressed many concerns, below are some remaining concerns that keep reviwers hesitant to recommend acceptance:
- Reviewer cKwh fails to be convinced about the use of positional-consistency. While acknowledging its precedence among other heuristics in prior work (as argued by the authors), they remain considering this design choice arbitrary.
- Reviewer GhxJ had concerns about the in-distribution results in Table 1, it was unclear why the SFT/DPO results are not better than other approaches using the same dataset. The authors made the effort running additional experiments that mixing the in-distribution data with new synthetic ID data and this achieves consistently better results. While applauding to that response, the reviewer believes further investigation is required to fully understand this, e.g., an ablation study on how much the performance improvement was from original data vs from the synthetic data.

I share some concerns with the reviewers as following:
- I partly agree with Reviewer cKwh that self-consistency and positional-consistency are a bit arbitrary. I acknowledge that both are necessary properties that a good evaluation path should have, and the authors adopt these properties by following prior works, but in my opinion consistency properties are far from suffiicient. Do the authors mention in the paper/response about the possibility that the LM fine-tuned with such data (with SFT and DPO) could fall into some local optima that overly-concern about consistency metrics but hurts on some other dimensions (especially on the semantics of an evaluation chain)?
- As also mentioned by Reviewer GhxJ, I have concerns about the diversity of the sampled trees. To my understanding, the diversity between branches could affect a lot the quality of the sampled data. As described in the paper (Section 4.3), the multiple criteria are generated with a brainstorming prompt, and the guidelines and judgements are sampled by playing with temperature and top-p parameters. I fail to find how they try to diversify the tree branches nor how to measure the diversity. In the authors' reponse to Reviewer GhxJ they say "only a minimal chance exists for the model to output semantically similar criteria" then refer us to Table 10 and 11 for examples. I fail to be convinced without seeing quantitative evidence; I fail to find qualitative evidence from Table 11 either (and actually the table does not show exmaple of guidelines?).

Overall this submission is around the borderline in terms of scores, the authors are doing a good job making their work stronger during the discussion period thank to some good suggestiong by the reviewers. I suggest the authors to continue improving their work especially in the line of better understanding its in-distribution behavior, as well as to measuring/encouraging diversity of the fine-tuning data.